# An Innovative Hybrid Heap-Based and Jellyfish Search Algorithm for Combined Heat and Power Economic Dispatch in Electrical Grids

Ahmed Ginidi [1], Abdallah Elsayed [2], Abdullah Shaheen [1], Ehab Elattar [3] and Ragab El-Sehiemy [4,*]

[1] Department of Electrical Engineering, Faculty of Engineering, Suez University, Suez 43533, Egypt; ahmed.ginidi@eng.suezuni.edu.eg (A.G.); abdullahshaheen2015@gmail.com (A.S.)

[2] Department of Electrical Engineering, Faculty of Engineering, Damietta University, Damietta 34517, Egypt; am.elsherif@yahoo.com

[3] Department of Electrical Engineering, College of Engineering, Taif University, Taif 21944, Saudi Arabia; e.elattar@tu.edu.sa

[4] Department of Electrical Engineering, Faculty of Engineering, Kafrelsheikh University, Kafrelsheikh 33516, Egypt

\* Correspondence: elsehiemy@eng.kfs.edu.eg

**Abstract:** This paper proposes a hybrid algorithm that combines two prominent nature-inspired meta-heuristic strategies to solve the combined heat and power (CHP) economic dispatch. In this line, an innovative hybrid heap-based and jellyfish search algorithm (HBJSA) is developed to enhance the performance of two recent algorithms: heap-based algorithm (HBA) and jellyfish search algorithm (JSA). The proposed hybrid HBJSA seeks to make use of the explorative features of HBA and the exploitative features of the JSA to overcome some of the problems found in their standard forms. The proposed hybrid HBJSA, HBA, and JSA are validated and statistically compared by attempting to solve a real-world optimization issue of the CHP economic dispatch. It aims to satisfy the power and heat demands and minimize the whole fuel cost (WFC) of the power and heat generation units. Additionally, a series of operational and electrical constraints such as non-convex feasible operating regions of CHP and valve-point effects of power-only plants, respectively, are considered in solving such a problem. The proposed hybrid HBJSA, HBA, and JSA are employed on two medium systems, which are 24-unit and 48-unit systems, and two large systems, which are 84- and 96-unit systems. The experimental results demonstrate that the proposed hybrid HBJSA outperforms the standard HBA and JSA and other reported techniques when handling the CHP economic dispatch. Otherwise, comparative analyses are carried out to demonstrate the suggested HBJSA's strong stability and robustness in determining the lowest minimum, average, and maximum WFC values compared to the HBA and JSA.

**Keywords:** heap-based algorithm; jellyfish search algorithm; economic dispatch; combined heat and power plants

## 1. Introduction

The energy supply in the globe is shifting toward high efficiency, sustainability, and low carbon content [1]. In conventional power units, a large amount of energy is wasted during the conversion of fossil fuels into electricity because of the low efficiency of these conventional plants. However, by utilizing the CHP economic dispatch, the whole fuel cost (WFC) may be reduced by 10–40%, energy efficiency can be increased to 90%, and greenhouse gases (GHG) can be reduced by roughly 13–18% [2]. The heat and electrical energy in the CHP system can be generated from a single source at the same time. The vital optimization challenge for the CHP economic dispatch is to find the minimum WFC of heat and power supply. There are several constraints that should be considered in the CHP economic dispatch, involving a load balance of the system, capacity limitations

of generation plants, the valve-point effect of thermal plants, and the heat and power mutual dependency provided by CHP. Two main categories of optimization approaches are explained to solve the CHP economic dispatch problem in recent research, comprising mathematical and heuristic optimization techniques [3,4].

One such task is the economic dispatch of the power system, which entails coordination, planning, and scheduling generators in an efficient manner. Due to the imposed equality and inequality restrictions, the economic dispatch issue exhibits nonlinear behavior. The economic dispatch problem has been highlighted as a multimodal optimization problem that will be difficult to tackle. Because actual issues are multimodal in nature, gradient methods are inapplicable [5]. In [6], an enhanced multi-objective particle swarm optimizer (MOPSO) model was used to manage a bi-objective dispatch framework in order to enhance the power quality and economic costs. In this study, a deep learning approach has been used to improve wind forecast accuracy where uncertainty analysis is a critical component of any assessment of a wind farm's long-term electricity output [7]. In [8], an improved antlion optimizer was presented to search for potential solutions for the economic dispatch issue in power systems with thermal units in order to minimize the generating fuel costs and guarantee that all restrictions are within functioning ranges. In [9], a modified crow search optimization was applied for solving the economic dispatch considering the environmental impacts and high-voltage direct current systems.

Added to that, the CHP economic dispatch has been solved throughout lots of conventional and mathematical approaches. In [10], a decentralized solution based on bender decomposition (BD) was performed for the optimal schedule of the CHP economic dispatch. The Lagrange relaxation (LR) and LR with surrogate sub-gradient (LRSS) have been employed in [11,12] with two levels to find out the optimal solution for studying the CHP economic dispatch. In [13], sequential quadratic programming (SQP) was combined with the LR method, where the LR technique was applied to the optimal CHP scheduling, and SQP was applied on a portion of the CHP problem to check the validity of the acquired operating point inside the trust region. In [14], the envelope-based branch and bound (EBB) approach was utilized for optimal planning of the CHP.

However, to deal with the non-convex objective function of the CHP economic dispatch and to overcome computational time efforts, heuristic approaches have been applied on the mentioned problem, such as the genetic algorithm (GA) [15], opposition teaching learning-based optimization (OTLBO) [16], differential evolution (DE) [17], multi-player harmony search (MPHS) algorithm [18], cuckoo search (CS) [19], and whale optimization algorithm (WOA) [20]. In [21], a greedy randomized adaptive search procedure (GRASP) method was hybridized with DE optimization and applied for the CHP economic dispatch to increase global search capacity while avoiding converging to local minima. In [22], an advanced mutation mechanism was involved in real coded GA and applied to the CHP economic dispatch for minimizing the operation cost, in order to enhance the convergence characteristics. In [23], an improved GA based on a new crossover and mutation was utilized to solve the CHP economic dispatch problem for handling constraints and applied to four cases for assessing the performance of the approach. In [24], a biogeography-based learning PSO (BLPSO) was carried out to improve the solution accuracy and overcome premature convergence where each particle utilized a migration operator to update itself depending on the best position of the whole particles. In addition, a multi-objective PSO has emerged with non-dominated sorting GA [25], and a modified version of shuffle frog leaping (MVSFL) algorithm [26] has been successfully employed on the CHP economic dispatch with limited small-scale applications, which are 5-unit and 7-unit systems.

The authors of [27] presented a combined optimization approach for power systems, which managed energy with power market and active microgrids in electric vehicle parking lots, diverse CHP economic dispatches, power and heat storage units, and distributed production. In [28], a Manta ray foraging optimizer (MRFO) was incorporated with adaptive constraint handling for solving the CHP economic dispatch, whereas the impact of the inclusion of wind power based on the MRFO was investigated in [29]. Moreover, a two-stage

mathematical programming has been proposed in [30] to deal with the nondifferentiable portion of valve-point loading influence and attain a convex operating zone in the CHP economic dispatch problem. In [31], the authors investigated the heat in power equipment and the availability of power flexibility in CHP technology from district heating networks.

Recently, two novel algorithms, heap-based algorithm (HBA) and jellyfish search algorithm (JSA), have been introduced to solve global optimization problems. Firstly, the HBA is a powerful metaheuristic optimization that is inspired from organization hierarchy created by Qamar Askari et al. [32]. Its simplicity and effectiveness enforce the research direction into its promising implementations in solving engineering problems. In [33], the HBA was efficiently utilized for parameter estimation of fuel cells, while it was applied for the CHP economic dispatch in [34] and optimal reactive power dispatch in [35]. Secondly, the standard JSA, inspired from jellyfish movements, was created by J.-S. Chou and D.-N. Truong in January 2021 [36]. In [37], the JSA was employed for a spectrum defragmentation algorithm in an elastic optical network. In [38], the JSA was utilized for efficient power system operation based on optimal power flow, whereas it was effectively applied in distribution networks to integrate distributed generators and the static volt-ampere reactive compensator [39]. In this paper, a novel hybrid heap-based and jellyfish search algorithm (HBJSA) is proposed, which combines the benefits of the standard HBA and standard JSA. Compared with the standard HBA and standard JSA, the proposed HBJSA uses an adjustment mechanism to support explorative and exploitative characteristics. The adjustment mechanism is constructed to boost the explorative features at the start of iterations by enhancing the generated solutions via HBA. Furthermore, towards the conclusion of iterations, it augments and enhances the exploitative features by growing the generated solutions via JSA. The efficiency of the HBA, JSA, and the proposed HBJSA is evaluated for solving the CHP economic dispatch by considering various constraints of heat production and power output balance.

The rest of this paper is structured as follows: the CHP economic dispatch problem is characterized in Section 2, whereas Section 3 includes a description of the standard HBA, the standard JSA, and the proposed hybrid HBJSA. Furthermore, Section 4 presents the outcomes of these algorithms and discussion for simulation, while a conclusion is presented in Section 5 of this work.

## 2. Problem Formulation

The general form of the CHP economic dispatch problem is described in Figure 1. This figure shows the single line diagram of the 24-unit test system for the CHP economic dispatch problem. As shown, different sources of the CHP, heat only, and power-only units supply power and heat are combined together to satisfy the power and heat demands. Heat production and power output balance means that the total power generation equals the total power load and the total heat generation equals the total heat load.

The objective function of the CHP economic dispatch problem can be illustrated as depicted in the following equation [2]:

$$Min\{WFC\} = Min\left\{ \sum_{i=1}^{Npp} C_i\left(P_i^{pp}\right) + \sum_{h=1}^{Nhp} C_h\left(H_h^{hp}\right) + \sum_{k=1}^{NCp} C_k\left(P_k^{Cp}, H_k^{Cp}\right) \right\}(\text{USD}/\text{h}) \quad (1)$$

The three terms of costs manifested in Equation (1) are explained in Equations (2)–(4) as in [20]. The cost function of a power-only plant involves quadratic and sinusoidal terms, where the sinusoidal term displays the valve-point impacts as signified in Equation (2). Furthermore, the heat-only cost is formulated in Equation (3), while the CHP cost function is represented in Equation (4).

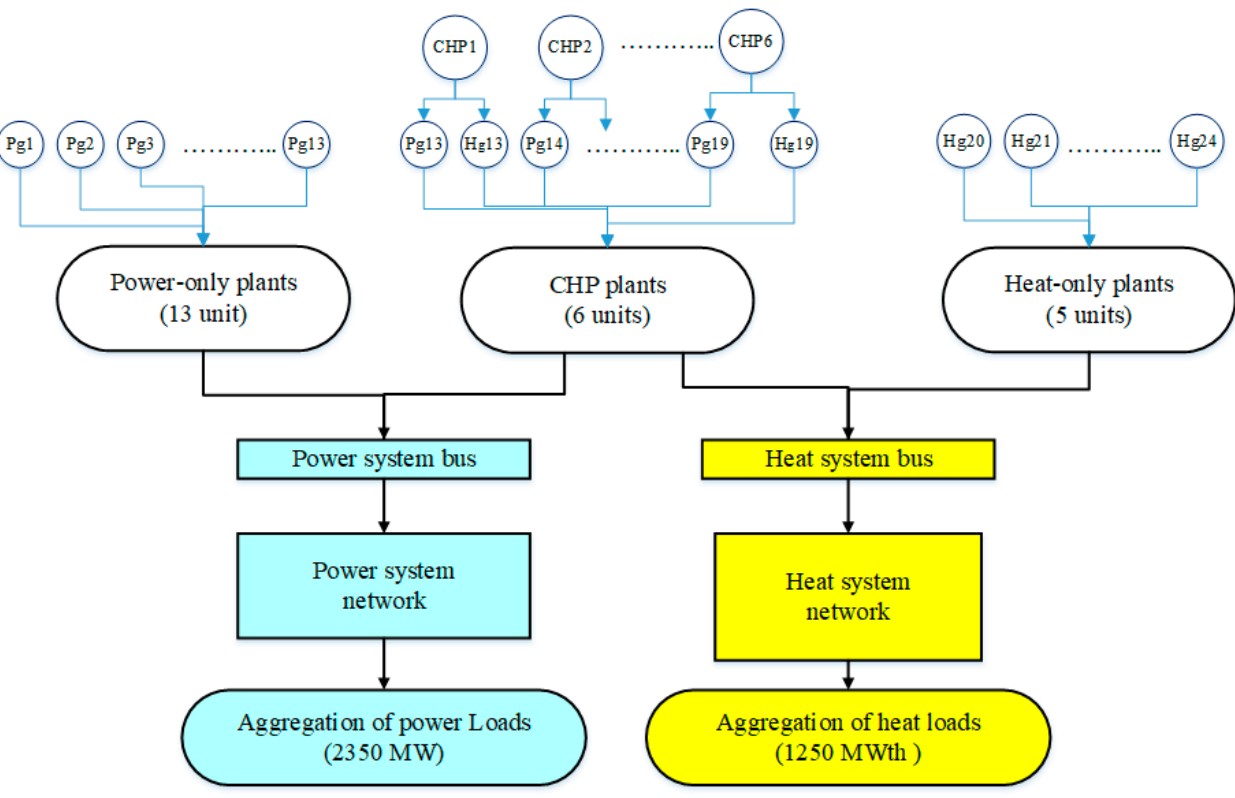

**Figure 1.** A single line diagram of the CHP economic dispatch problem considering the 24-unit test system.

$$C_i(P_i^{pp}) = a_i(P_i^{pp})^2 + b_i P_i^{pp} + c_i + \left| \lambda_i \sin(\rho_i(P_i^{pp\min} - P_i^{pp})) \right| \text{ (USD/h)} \tag{2}$$

$$C_j(H_j^{hp}) = a_j(H_j^{hp})^2 + b_j P_j^{pp} + c_j \text{ (USD/h)} \tag{3}$$

$$C_k(P_k^{cp}, H_k^{cp}) = a_k(P_k^{cp})^2 + b_k P_k^{pp} + c_k + d_k(H_k^{cp})^2 + e_k H_k^{cp} + f_k H_k^{cp} P_k^{cp} \text{(USD/h)} \tag{4}$$

Diverse constraints for feasible solutions are illustrated for the CHP economic dispatch problem as follows:

$$\sum_{i=1}^{N_{pp}} P_i^{pp} + \sum_{j=1}^{N_{cp}} P_j^{cp} = P_d \tag{5}$$

$$\sum_{j=1}^{N_{cp}} H_j^{cp} + \sum_{k=1}^{N_{hp}} H_k^{hp} = H_d, \tag{6}$$

Furthermore, power-only and heat-only capacity limits are exposed in Equation (7) and Equation (8), respectively. In addition to that, capacity limits of CHP are designated in Equations (9) and (10).

$$P_i^{pp\min} \le P_i^{pp} \le P_i^{pp\max} \quad i = 1, \dots, N_{pp}, \tag{7}$$

$$H_j^{hp\min} \le H_j^{hp} \le H_j^{hp\max} \quad i = 1, \dots, N_{hp}, \tag{8}$$

$$P_k^{cp\min}(H_k^{cp}) \le P_k^{cp} \le P_k^{cp\max}(H_k^{cp}) \quad k = 1, \dots, N_{cp}, \tag{9}$$

$$H_k^{cp\min}(P_k^{cp}) \le H_k^{cp} \le H_k^{cp\max}(P_k^{cp}) \quad k = 1, \dots, N_{cp}, \tag{10}$$

In the above constraints, Equations (5) and (6) demonstrate the power generated and the power demand balance and the heat generated and the demand balance, respectively.

### 3. Hybrid HBJSA for CHP Economic Dispatch Problem

*3.1. Standard HBA*

The standard HBA concept is based on the corporate rank hierarchy (CRH), which states that a team can arrange itself in a hierarchy to fulfill organizational goals [32]. The HBA is classified into three levels: interaction among subordinates, self-contribution of employees and their immediate supervisor, and interaction among colleagues.

In the CRH model, the population is manifested by the full CRH, whereas the heap node is manifested by the search agent. The search agent's fitness is the master of the heap node, and the population index of the search agent is the value of the heap node. The agent position of each search can be updated as:

$$x_i^k(t+1) = B^k + \gamma(2r-1)\left|B^k - x_i^k(t)\right| \tag{11}$$

The *k*th component of λ vector $\vec{\lambda}$ is represented by:

$$\lambda^k = 2r - 1 \tag{12}$$

γ is computed as follow:

$$\gamma = \left|2 - \frac{\left(t \bmod \frac{t}{C}\right)}{\frac{t}{4C}}\right| \tag{13}$$

The parameter (*C*) in Equation (14) controls the variation. However, this parameter will complete in *T* iterations as follows:

$$C = T^{\max}/25 \tag{14}$$

Added to that, the interaction between colleagues is modeled. As manifested in Equation (15), the position of each agent $(\vec{x_i})$ is updated by its arbitrarily selected colleague $\vec{S_r}$:

$$x_i^k(t+1) = \begin{cases} S_r^k + \gamma\lambda^k\left|S_r^k - x_i^k(t)\right|, & f(\vec{S_r}) < f(\vec{x_i}(t)) \\ x_i^k + \gamma\lambda^k\left|S_r^k - x_i^k(t)\right|, & f(\vec{S_r}) \geq f(\vec{x_i}(t)) \end{cases} \tag{15}$$

where the fitness of the search agent can be represented by *f*.

Additionally modeled is the self-contribution of each employee, where the position of each agent is updated in this level according to the following equation:

$$x_i^k(t+1) = x_i^k(t) \tag{16}$$

Finally, the position updating equations have been emerged. The roulette wheel probabilities, $p_1$, $p_2$, and $p_3$, are selected to balance the exploration and exploitation processes. The search agent updates its position using Equation (16). Selecting the proportion $p_1$ is carried out by using Equation (17) as:

$$p_1 = 1 - \frac{t}{T^{max}} \tag{17}$$

The search agent updates its position using Equation (11). Selecting the proportion $p_2$ is carried by using Equation (18) as:

$$p_2 = p_1 + \frac{1 - p_1}{2} \tag{18}$$

The search agent updates its position using Equation (17). Selecting the proportion $p_3$ is carried out by using Equation (19) as:

$$p_3 = p_2 + \frac{1 - p_1}{2} = 1 \qquad (19)$$

Hence, the general positions' updating mechanism of the HBA is formulated as in Equation (20):

$$x_i^k(t+1) = \begin{cases} x_i^k(t), & p \le p_1 \\ B^k + \gamma \lambda^k \left| B^k - x_i^k(t) \right|, & p_1 < p < p_2 \\ S_r^k + \gamma \lambda^k \left| S_r^k - x_i^k(t) \right|, & p_2 < p \le p_3 \text{ and } f(\vec{S_r}) < f(\vec{x_i}(t)) \\ x_r^k + \gamma \lambda^k \left| S_r^k - x_i^k(t) \right|, & p_2 < p \le p_3 \text{ and } f(\vec{S_r}) \ge f(\vec{x_i}(t)) \end{cases} \qquad (20)$$

The main steps of the proposed HBA are depicted in Figure 2.

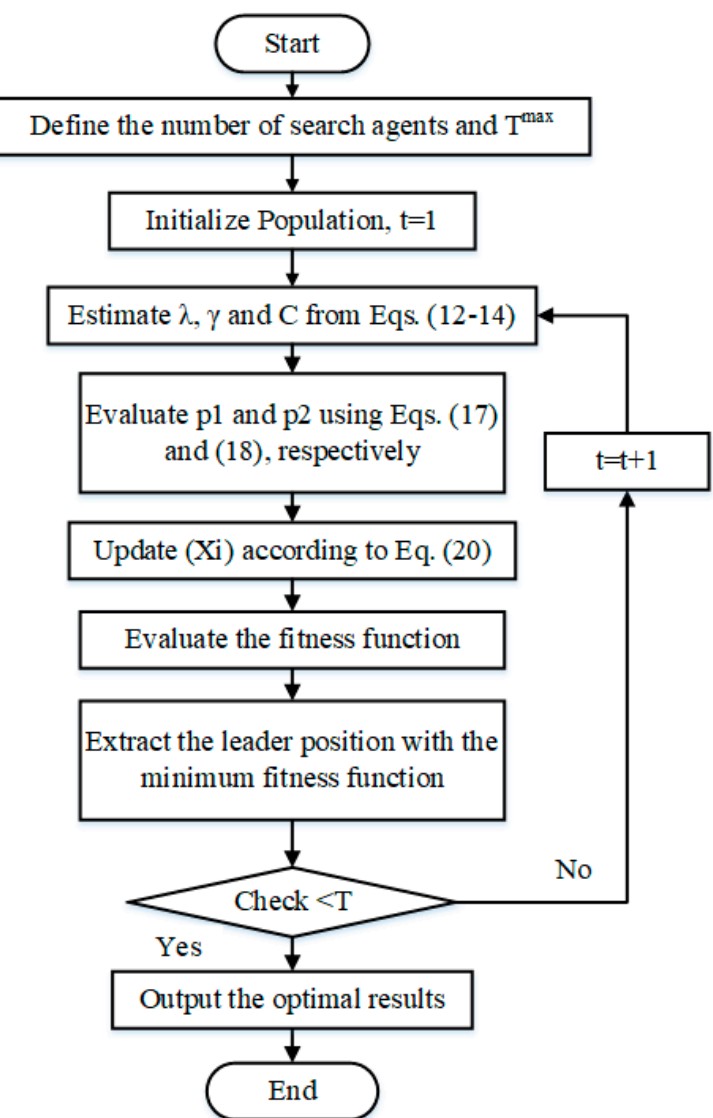

**Figure 2.** Flowchart of the HBA.

### 3.2. Standard JSA

The JSA is inspired by the jellyfish movements whether they move in the ocean current or within their swarm [36]. The jellyfish population can be mathematically modeled as:

$$X_i(t+1) = 4P_0(1 - X_i), \quad 0 \le P_0 \le 1 \tag{21}$$

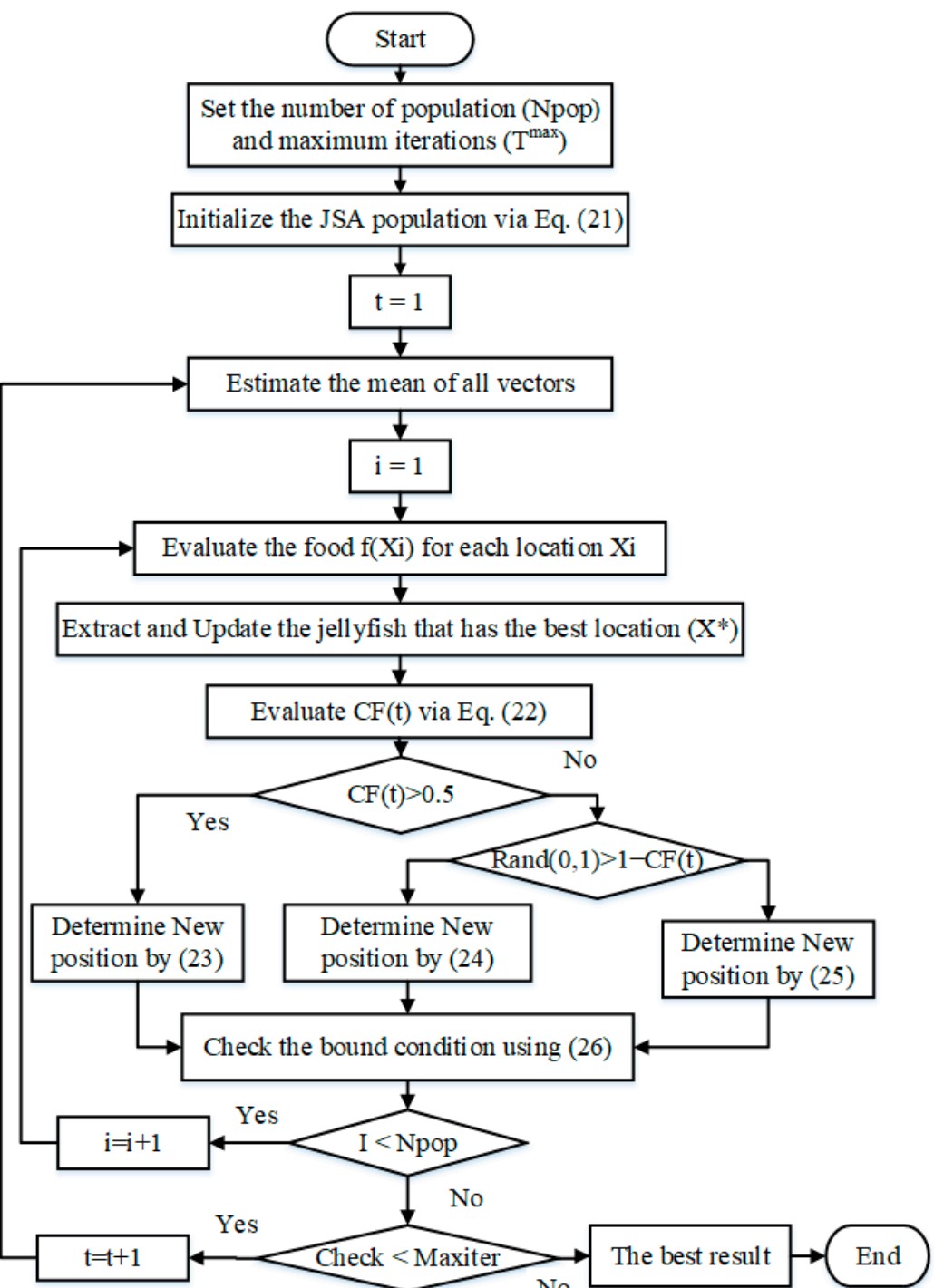

**Figure 3.** Flowchart of the JSA.

The time control function $CF(t)$ value is assessed as described in Equation (22), and it is varied from 0 to 1 over time:

$$CF(t) = \left| \left( 1 - \frac{t}{T^{\max}} \right) \times (2 \times rand(0,1) - 1) \right| \tag{22}$$

If the $CF$ is greater than the constant $CO_o$ (to be 0.5), the new location of each jellyfish can be formulated as demonstrated in Equation (23)

$$X_i(t+1) = R \times (X^* - 3 \times R \times \mu) + X_i(t) \tag{23}$$

If $CF$ value is more than $CO_o$, each jellyfish location is updated depending on the movement within the swarm, as clarified in Equations (24) and (25).

$$X_i(t+1) = 0.1 \times R \times (U_b - L_b) + X_i(t) \tag{24}$$

$$X_i(t+1) = \begin{cases} X_i(t) + R \times (X_j(t) - X_i(t)) & if \ f(X_i) \geq f(X_j) \\ X_i(t) + R \times (X_i(t) - X_j(t)) & if \ f(X_i) < f(X_j) \end{cases} \tag{25}$$

As soon as a jellyfish moves at the back of the search zone boundaries, it will go back, as is demonstrated in Equation (26), to the reverse boundary.

$$\begin{cases} X'_{i,d} = (X_{i,d} - U_{b,d}) + L_b(d) & if \quad X_{i,d} > U_{b,d} \\ X'_{i,d} = (X_{i,d} - L_{b,d}) + U_b(d) & if \quad X_{i,d} < L_{b,d} \end{cases} \tag{26}$$

where $X_{i,d}$ expresses the $i$th jellyfish location in $d$th dimension. The main steps of the JSA are depicted in Figure 3.

### 3.3. Proposed Hybrid HBJSA

In this sub-section, a hybrid HBJSA is proposed to combine the benefits of the standard HBA and standard JSA. Compared with standard HBA and standard JSA, the proposed HBJSA employs an adjustment mechanism to support the explorative and exploitative characteristics. This mechanism is constructed to boost the explorative feature at the start of iterations by enhancing the generated solutions via HBA. Furthermore, toward the conclusion of iterations, it augments and enhances the exploitative feature by growing the generated solutions via JSA. The adjustment mechanism is executed by employing an adaptive coefficient ($\varphi$) designed as follows:

$$\varphi = \frac{t}{2 \times T^{\max}} \tag{27}$$

From this equation, the coefficient ($\varphi$) is correlated positively with the number of iterations increases until it reaches 0.5 at the highest quantity of iterations. The more the value of the coefficient ($\varphi$) increases, increasing of the generated solutions via JSA will be updated by Equation (28) as follows:

$$x_i^k(t+1) = R \times (Leader^k - 3 \times R \times \mu) + x_i^k(t) \tag{28}$$

where *Leader* is the leader position of the search agents, which achieves the minimum fitness value.

Another point of view for handling the CHP economic dispatch problem, the objective function in Equation (1) is updated to incorporate penalized terms of the power and heat units constraints as follows:

$$OF = WFC + Pen_v \sum_{j=1}^{N_c} B_v \left( P_j^C(H_j^C) - P_j^{C\,Limit}(H_j^C) \right) \tag{29}$$

where the term $(P_j^{CLimit} (H_j^C))$ is the power limit to the CHP j heating output; $\psi_v$ is a penalized coefficient for CHP operating violating; $Bv$ equals 1 when there is violation or 0 when there is not. Accordingly, the farthest violated operating point will have a greater penalty.

Figure 4 illustrates the main steps of the proposed hybrid HBJSA for handling the CHP economic dispatch problem. For more information about the proposed HBJSA, the main steps can be summarized as follows:

- Define the parameters of HBJSA.
- Randomly initialize the control parameters that involve the output of power and heat of the committed units and keep it within the accepted boundaries. They are checked versus its acceptable bounds' mechanism. Both units began inside their respective limitations, and if either of them is violated throughout the repetitions, it is reset to the nearest limit.
- Evaluate the fitness function of the CHP problem that minimizes the overall cost using Equation (28).

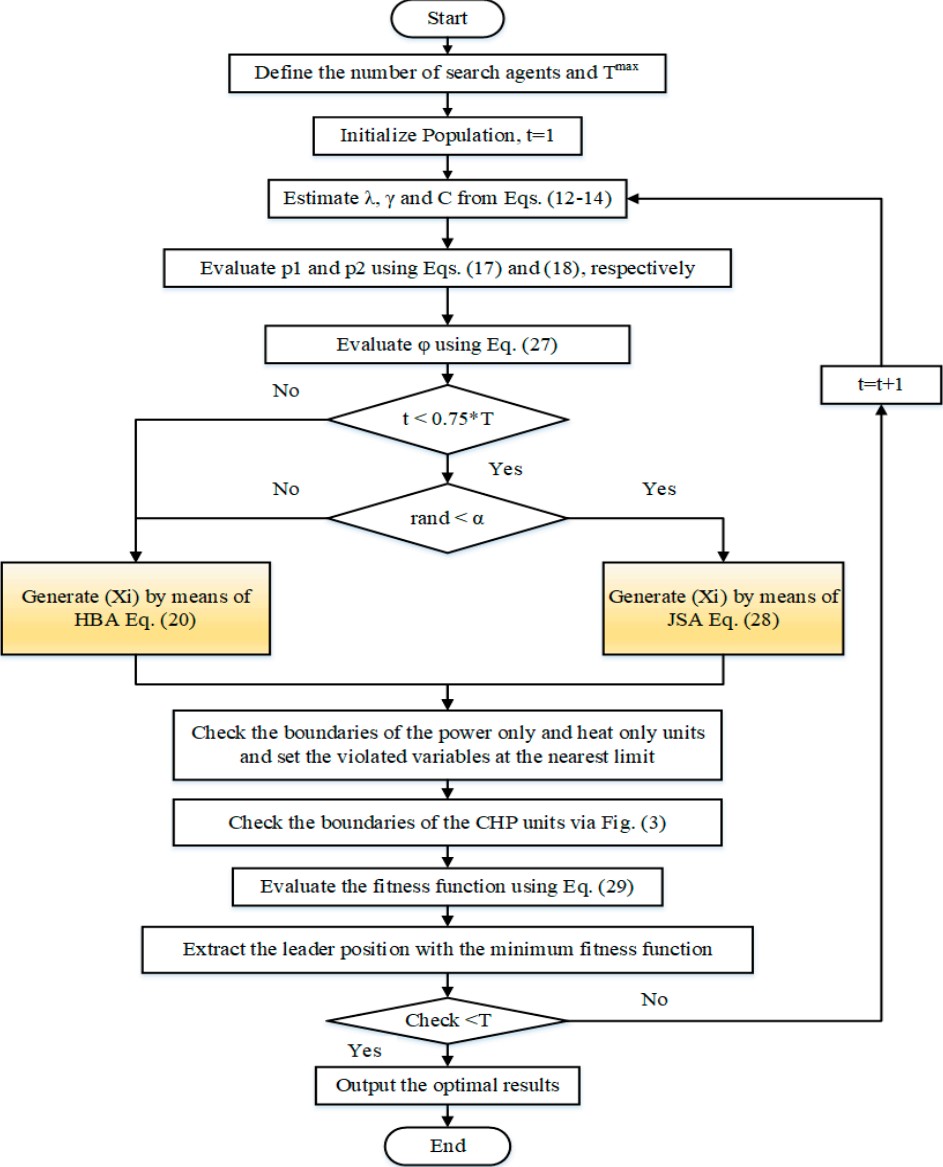

**Figure 4.** Flowchart of the proposed HBJSA.

Figure 3 shows the second type of mutually dependent CHP unit. They are dealt with as the penalty function that is added in the considered fitness function (OF) in Equation (29). As shown in Figure 3, when an operational point is inside the limits, it has a *Bv* value of zero, while the infeasible locations have a *Bv* value of one. On the other side, the greater the penalty amount, the farther the infeasible point is from the nearest border.

As a result, the proposed hybrid HBJSA has a greater capacity for looking for suitable locations. Furthermore, a stopping condition is used in which the ideal result is attained if the maximum number of iterations is reached. HBA penalizes infeasible solutions to varying degrees based on their distance from the next feasible point.

## 4. Simulation Results

The proposed HBJSA, the standard HBA, and JSA are employed on four test systems. The first two test systems are medium-scale 24-unit and 48-unit systems, whereas the second two test systems are large-scale 84-unit and 96-unit test systems. The number of iterations (T) and population size ($n_{pop}$), which are the main two parameters of the standard HBA, the standard JSA, and the proposed HBJSA account for 3000 and 100, respectively, for all systems. MatlabR2017b is utilized to carry out the simulations using CPU (2.5 GHz) Intel(R)-Core (TM) i7-7200U and 8 GB of RAM.

### 4.1. Simulation Results of the 24-Unit Test System

The data for the obtained system are mentioned in [40], which illustrates that 2350 MW and 1250 MWth are the load demand and heat demand, respectively, and it has five heat units, 13 thermal units, and six CHP units. The HBA, JSA, and proposed HBJSA are applied on this test system, and the corresponding MW, MWth for each unit, and WFC are demonstrated in Table 1. It can be manifested that the proposed HBJSA provides the optimal solution for WFC minimization, which accounts for USD 57,968.5399, while the standard HBA and the standard JSA account for USD 57,994.51 and USD 58,739.5241, respectively.

Moreover, convergence characteristics of the proposed HBJSA versus the standard HBA and the standard JSA for the 24-unit test system of the CHP economic dispatch problem are depicted in Figure 5. It is seen from that figure that the proposed hybrid HBJSA is capable of improving the solution quality compared to the standard HBA and the standard JSA. At the last 400 iterations, the proposed hybrid HBJSA provides a higher exploitative feature and, finally, reaches the lowest WFC of USD 57,968.5399. Additionally, the standard HBA, the standard JSA, and the proposed HBJSA effectively achieve all constraints with 100% accuracy, as illustrated in Table 1.

In addition, a comparison between the HBA, JSA, and the proposed HBJSA is conducted in Table 2 for the 24-unit system of CHP economic dispatch with respect to reported techniques such as PSO [40], time-varying acceleration coefficients-PSO (TVAC-PSO) [40], group search optimization (GSO) [41], and improved GSO (IGSO) [42], MRFO [28], and supply demand algorithm (SDA) [34]. In this table, ranking order is evaluated in ascending order based on the minimum WFC. From this table, the proposed hybrid HBJSA achieves the first rank with the lowest WFC. On the other side, the standard HBA occupies the second rank, while the standard JSA occupies the last rank. This table demonstrates that the proposed HBJSA overwhelmed the standard HBA, the standard JSA, and the reported recent techniques for achieving minimum WFC.

**Table 1.** Comparison between HBA, JSA, and the proposed HBJSA for the 24-unit test system of CHP economic dispatch problem.

| Unit | JSA | HBA | HBJSA |
|------|-----|-----|-------|
| P 1 | 449.27558 | 538.55874 | 448.81809 |
| P 2 | 149.67888 | 300.2175 | 299.21886 |
| P 3 | 202.56201 | 301.08256 | 300.72118 |
| P 4 | 109.86032 | 159.77793 | 60.109633 |
| P 5 | 109.93156 | 63.21736 | 159.74512 |
| P 6 | 159.73642 | 60.688903 | 159.77696 |
| P 7 | 160.00828 | 160.20653 | 159.77184 |
| P 8 | 159.74295 | 111.5383 | 60.000007 |
| P 9 | 109.83449 | 161.25396 | 159.75102 |
| P 10 | 77.389984 | 40 | 77.411833 |
| P 11 | 77.406979 | 40.000266 | 40.001098 |
| P 12 | 92.367201 | 55.657936 | 55.008621 |
| P 13 | 92.395174 | 55.284533 | 55.661102 |
| P 14 | 115.82103 | 87.944171 | 85.844198 |
| P 15 | 40.964462 | 41.266255 | 42.751997 |
| P 16 | 114.87371 | 84.034893 | 95.888699 |
| P 17 | 69.301243 | 43.143672 | 44.468374 |
| P 18 | 10.133787 | 11.08247 | 10.046228 |
| P 19 | 48.715961 | 35.04403 | 35.005125 |
| H 14 | 124.27642 | 108.69733 | 107.49154 |
| H 15 | 75.711188 | 76.092716 | 77.376451 |
| H 16 | 123.80751 | 106.47628 | 113.15577 |
| H 17 | 100.29358 | 77.714606 | 78.85075 |
| H 18 | 40.003606 | 40.464341 | 40.020006 |
| H 19 | 26.210361 | 20.020468 | 20.001274 |
| H 20 | 399.97537 | 460.53782 | 453.10933 |
| H 21 | 59.9258 | 60 | 60 |
| H 22 | 59.902038 | 60 | 59.998827 |
| H 23 | 119.91378 | 119.99644 | 119.99649 |
| H 24 | 119.98035 | 120 | 119.99957 |
| Sum (Pg) | 2350.0000 | 2350.0000 | 2350.0000 |
| Sum (Hg) | 1250.0000 | 1250.0000 | 1250.0000 |
| WFC | 58,739.5241 | 57,994.5150 | 57,968.5399 |

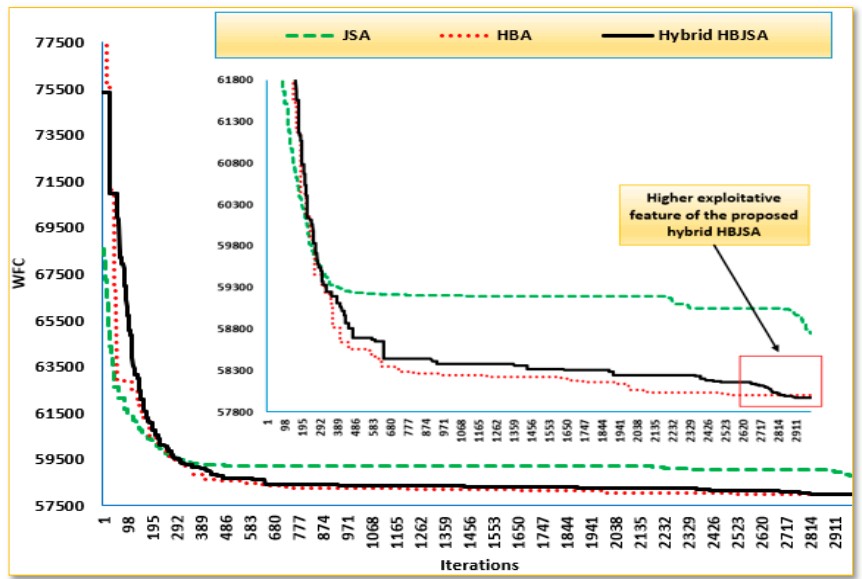

**Figure 5.** Convergence characteristics of the proposed HBJSA versus the HBA and JSA for the 24-unit system of CHP economic dispatch.

**Table 2.** Comparison between HBA, JSA, and HBJSA with respect to reported techniques for the 24-unit system of CHP economic dispatch.

| Method | Sum (Pg) | Sum (Hg) | WFC | Rank |
|---|---|---|---|---|
| GSO [41] | 2350 | 1250 | 58,225.745 | 7 |
| IGSO [41] | 2350 | 1250 | 58,049.01 | 3 |
| PSO [40] | 2349.9 | 1250 | 59,736.26 | 8 |
| TVAC-PSO [40] | 2350.0002 | 1250 | 58,122.74 | 5 |
| MRFO [28] | 2350 | 1250 | 58,173.93 | 6 |
| SDA [34] | 2350 | 1250 | 58,061.477 | 4 |
| JSA | 2350 | 1250 | 58,739.5241 | 9 |
| HBA | 2350 | 1250 | 57,994.515 | 2 |
| Proposed hybrid HBJSA | 2350 | 1250 | 57968.54 | 1 |

*4.2. Simulation Results of the 48-Unit Test System*

**Table 3.** Comparison between HBA, JSA, and the proposed HBJSA for the 48-unit test system of CHP economic dispatch problem.

| Output | HBA | JSA | HBJSA | Output | HBA | JSA | HBJSA |
|---|---|---|---|---|---|---|---|
| P 1 | 628.31969 | 538.55876 | 628.31847 | P 33 | 89.741692 | 93.956738 | 92.056329 |
| P 2 | 301.64745 | 166.31944 | 298.49068 | P 34 | 45.760605 | 63.520906 | 44.496057 |
| P 3 | 297.56062 | 299.2078 | 298.71921 | P 35 | 93.563713 | 98.369246 | 95.721078 |
| P 4 | 60.025209 | 109.86692 | 162.17614 | P 36 | 54.117683 | 64.357184 | 46.047531 |
| P 5 | 60.891955 | 159.73312 | 60 | P 37 | 10.89958 | 16.476816 | 11.179988 |
| P 6 | 115.51041 | 109.86683 | 60.187387 | P 38 | 35 | 47.149532 | 36.636756 |
| P 7 | 60 | 109.87111 | 60.000003 | H 27 | 109.52988 | 130.11379 | 112.92198 |
| P 8 | 112.29664 | 109.86759 | 61.265074 | H 28 | 77.887623 | 76.952262 | 80.359207 |
| P 9 | 163.79819 | 109.86802 | 60.012628 | H 29 | 106.94451 | 118.12661 | 112.31166 |
| P 10 | 84.337266 | 77.40216 | 40.077792 | H 30 | 79.211207 | 102.05398 | 78.341268 |
| P 11 | 44.380157 | 77.405413 | 77.401957 | H 31 | 42.302628 | 42.775213 | 40.213152 |
| P 12 | 92.442285 | 92.402577 | 92.236557 | H 32 | 21.949409 | 24.616716 | 20.01776 |
| P 13 | 92.547707 | 92.400831 | 55.000005 | H 33 | 108.80007 | 112.07165 | 111.00119 |
| P 14 | 538.58181 | 359.03989 | 628.66001 | H 34 | 79.972231 | 95.305162 | 78.407382 |
| P 15 | 150.5131 | 149.62114 | 299.67572 | H 35 | 111.85121 | 114.54771 | 112.98988 |
| P 16 | 302.48049 | 299.20217 | 224.39881 | H 36 | 87.186384 | 96.027118 | 80.220448 |
| P 17 | 159.0721 | 109.86997 | 160.3726 | H 37 | 40.316118 | 42.776163 | 40.506134 |
| P 18 | 159.48447 | 110.38702 | 110.74399 | H 38 | 19.999898 | 25.522939 | 20.698728 |
| P 19 | 60.042597 | 109.86649 | 159.34188 | H 39 | 442.02953 | 399.58888 | 445.84515 |
| P 20 | 60.163088 | 109.86727 | 60 | H 40 | 59.963929 | 59.999977 | 60 |
| P 21 | 160.27015 | 109.8666 | 60.001242 | H 41 | 60 | 59.999861 | 59.99996 |
| P 22 | 158.67694 | 159.73453 | 162.0577 | H 42 | 119.87507 | 119.99992 | 120 |
| P 23 | 40 | 77.402895 | 78.899748 | H 43 | 119.99995 | 119.99993 | 119.99952 |
| P 24 | 40.430825 | 77.408107 | 40 | H 44 | 452.35143 | 399.52269 | 446.16766 |
| P 25 | 55.283719 | 92.400956 | 55 | H 45 | 59.829479 | 59.999808 | 59.99892 |
| P 26 | 55.000343 | 92.649622 | 55.003382 | H 46 | 59.999794 | 59.999907 | 60 |
| P 27 | 89.427389 | 126.10612 | 95.4717 | H 47 | 119.99989 | 119.99982 | 120 |
| P 28 | 43.348156 | 42.260587 | 46.55662 | H 48 | 119.99975 | 119.9999 | 120 |
| P 29 | 84.844588 | 104.74609 | 94.384639 | Sum (Pg) | 4700 | 4700 | 4700 |
| P 30 | 44.879351 | 71.338822 | 43.869636 | Sum (Hg) | 2500 | 2500 | 2500 |
| P 31 | 15.371858 | 16.474563 | 10.500572 | WFC (USD) | 116,439.96 | 117,365.09 | 116,140.34 |
| P 32 | 39.288183 | 45.156162 | 35.038114 | | | | |

The data for the obtained system are mentioned in [40], which illustrates that 4700 MW and 2500 MWth are the load demand and heat demand, respectively, and it has 10 heat units, 26 thermal units, and 12 CHP. The HBA, JSA, and proposed HBJSA are applied to this test system, and the corresponding MW, MWth for each unit, and WFC are demonstrated in Table 3. It can be manifested that the proposed HBJSA provides the optimal solution for WFC minimization, which accounts for USD 116,140.34, while the standard HBA and the standard JSA account for USD 116,439.96 and USD 117,365.09, respectively.

Moreover, convergence characteristics of the proposed HBJSA versus the standard HBA and the standard JSA for the 48-unit test system of the CHP economic dispatch problem are depicted in Figure 6. It is seen from that figure that the proposed hybrid HBJSA is capable of improving the solution quality compared to the standard HBA and the standard JSA. After 900 iterations, the suggested hybrid HBJSA delivers more exploitative features and ultimately achieves the lowest WFC of USD 116,140.34. Additionally, the standard HBA, the standard JSA, and the proposed HBJSA effectively achieve the power and heat balance constraints with 100% accuracy, as illustrated in Table 3.

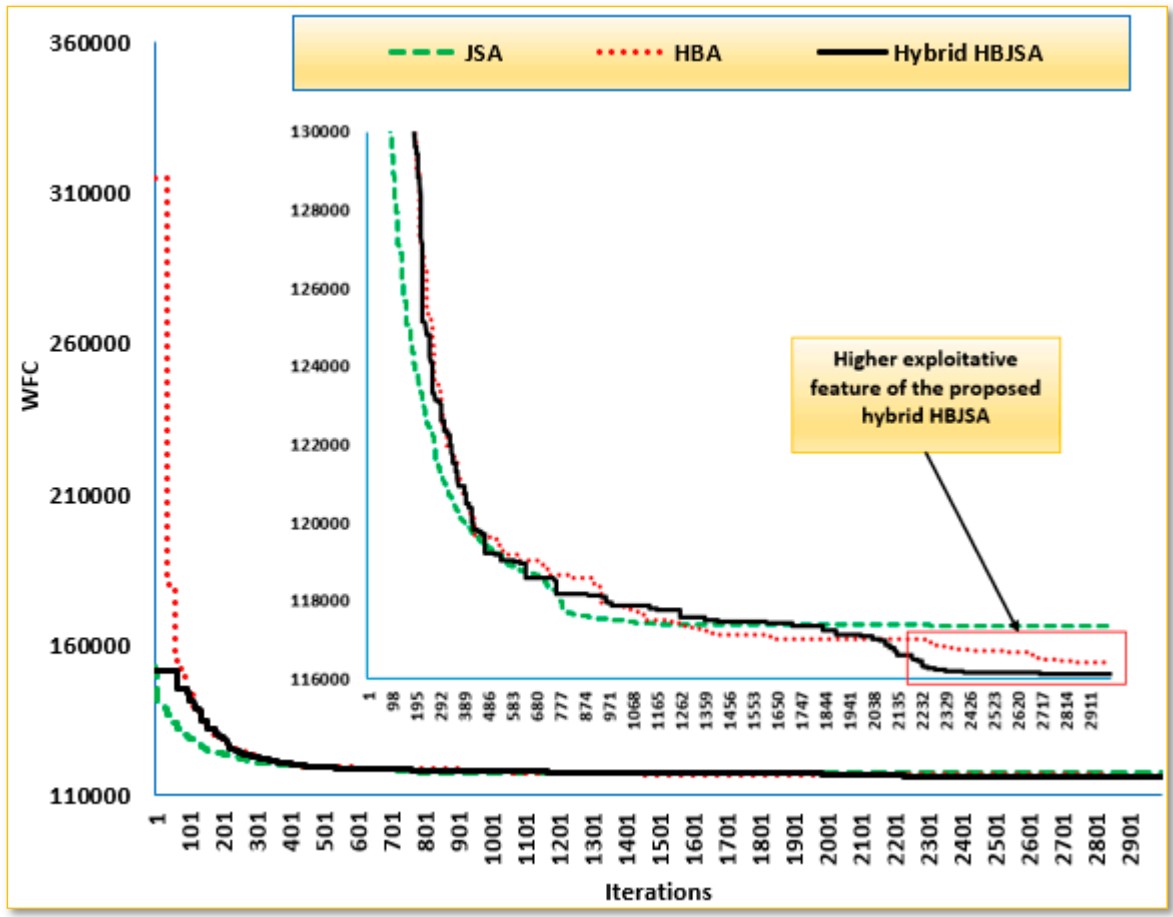

**Figure 6.** Convergence characteristics of the proposed HBJSA versus the HBA and JSA for the 48-unit system of CHP economic dispatch.

In addition to this, a comparison between the HBA, JSA, and proposed HBJSA is conducted in Table 4 for the 48-unit system of CHP economic dispatch with respect to other reported techniques such as MRFO [28], SDA [34] TVAC-PSO [40], CPSO [40], GSO [43], modified PSO [44], OTLBO [16], and MGSO [43] and gravitational search algorithm (GSA) [45]. Additionally, crow search algorithm (CSA) [46], grey wolf algorithm (GWA) [47], salp swarm algorithm (SSA) [48], multi-verse algorithm (MVA) [49], DE [50], MPA [51–53], civilized swarm optimization CSO [54] and Powell's pattern search PPS [54] are applied to the CHP economic dispatch for this system.

**Table 4.** Comparison between HBA, JSA, and HBJSA with respect to reported techniques for the 48-unit system of CHP economic dispatch.

| Optimizer | WFC (USD) | Rank | Optimizer | WFC (USD) | Rank |
|---|---|---|---|---|---|
| SDA [34] | 116,620.61 | 7 | Modified PSO [44] | 116,465.54 | 3 |
| MRFO [28] | 117,336.9 | 9 | TVAC-PSO [40] | 118,962.54 | 14 |
| CSA [28] | 122,953.5 | 20 | GSA [45] | 119,775.9 | 15 |
| GWA [28] | 122,583.3 | 19 | MPA [42] | 116,860.6 | 8 |
| SSA [48] | 120,174.1 | 16 | CSO and PPS [54] | 117,367.09 | 12 |
| MVA [28] | 117,657.9 | 13 | MGSO [43] | 117,366.09 | 11 |
| DE [28] | 120,482.7 | 17 | HBA | 116,439.96 | 2 |
| GSO [43] | 116,578.475 | 6 | JSA | 117,365.09 | 10 |
| OTLBO [16] | 116,579.24 | 4 | Proposed hybrid HBJSA | 116,331.21 | 1 |
| PSO [40] | 120,918.92 | 18 | | | |

In this table, ranking order is evaluated in ascending order based on the minimum WFC. From this table, the proposed hybrid HBJSA achieves the first rank with the lowest WFC. On the other side, the standard HBA occupies the second rank, while the standard JSA occupies the tenth rank. This table demonstrates that the proposed HBJSA overwhelmed the standard HBA, the standard JSA, and reported recent techniques for achieving minimum WFC.

*4.3. Simulation Results of the 84-Unit Test System*

The data for the tested system are mentioned in [20]. The power and heat demands equal 12,700 MW and 5000 MWth, respectively, and it has 20 heat units, 40 thermal units, and 24 CHP. The HBA, JSA, and proposed HBJSA are investigated to this test system, and the corresponding MW, MWth for each unit, and WFC are demonstrated in Table 5. It can be manifested that the proposed HBJSA provides the optimal solution for WFC minimization, which accounts for USD 288,820.7, while the standard HBA and the standard JSA account for USD 289,822.4 and USD 290,323.8, respectively.

Moreover, convergence characteristics of the proposed HBJSA are depicted in Figure 7 versus the standard HBA and the standard JSA for the 84-unit test system of the CHP economic dispatch problem. It is seen from that figure that the proposed hybrid HBJSA is capable of improving the solution quality compared with HBA and JSA. At the last 950 iterations, the proposed hybrid HBJSA provides a higher exploitative feature and, finally, reaches the lowest WFC of USD 288,820.7. Additionally, the standard HBA, the standard JSA, and the proposed HBJSA effectively achieve all constraints with 100% accuracy, as illustrated in Table 5.

In addition, a comparison study between the HBA, JSA, and proposed HBJSA is conducted in Table 6 for the 84-unit system of CHP economic dispatch with respect to reported techniques such as WOA [20], MRFO [28], marine predators algorithm (MPA) [42], improved MPA (IMPA) [42], and SDA [34]. In this table, ranking order is evaluated in ascending order based on the minimum WFC. From this table, the proposed hybrid HBJSA achieves the first rank with the lowest WFC. On the other side, the standard HBA occupies the second rank while the standard JSA occupies the fifth rank. This table demonstrates that the proposed HBJSA overwhelmed the standard HBA, the standard JSA, and reported recent techniques for achieving minimum WFC.

**Table 5.** Comparison between HBA, JSA, and the proposed HBJSA for the 84-unit test system of CHP economic dispatch problem. (a) Power outputs from power only and CHP units. (b) Heat outputs from CHP and heat-only units.

| | | | | (a) | | | |
|---|---|---|---|---|---|---|---|
| Unit | HBA | JSA | HBJSA | Unit | HBA | JSA | HBJSA |
| Pg1 | 114 | 111.1626385 | 113.9718986 | Pg33 | 181.7205 | 159.7499057 | 185.8014921 |
| Pg2 | 113.11556 | 112.5520392 | 112.0171823 | Pg34 | 199.99998 | 199.9449699 | 199.988706 |
| Pg3 | 103.83725 | 107.7765112 | 98.99471338 | Pg35 | 182.91598 | 199.6705009 | 181.7649853 |
| Pg4 | 184.80695 | 179.7375644 | 179.6906294 | Pg36 | 200 | 199.991963 | 199.9921324 |
| Pg5 | 89.505208 | 87.78776144 | 94.40847581 | Pg37 | 109.99936 | 89.86684403 | 109.9315908 |
| Pg6 | 106.64827 | 139.9948775 | 138.9511133 | Pg38 | 110 | 109.9917663 | 104.3481595 |
| Pg7 | 256.25545 | 266.7806527 | 260.1628409 | Pg39 | 89.839986 | 94.16960534 | 109.9999047 |
| Pg8 | 297.05131 | 290.8186132 | 299.6320704 | Pg40 | 550 | 511.3076161 | 517.5996621 |
| Pg9 | 299.99539 | 284.6025908 | 288.5380215 | Pg41 | 126.91167 | 132.3982552 | 110.3909623 |
| Pg10 | 130 | 207.4135385 | 130 | Pg42 | 126.65154 | 144.4049507 | 136.7428407 |
| Pg11 | 169.30903 | 243.5827337 | 242.0952346 | Pg43 | 115.3838 | 88.99398926 | 134.0103426 |
| Pg12 | 306.09411 | 318.3996454 | 168.8036849 | Pg44 | 133.27883 | 105.2551322 | 117.7366501 |
| Pg13 | 394.50082 | 304.5241963 | 394.2779192 | Pg45 | 42.80197 | 93.63589835 | 49.34154951 |
| Pg14 | 393.7356 | 304.5184491 | 394.2937412 | Pg46 | 43.679119 | 49.07352085 | 43.51922726 |
| Pg15 | 305.53666 | 394.322214 | 396.1931056 | Pg47 | 77.280238 | 76.78719074 | 59.37920101 |
| Pg16 | 394.45006 | 304.521412 | 394.3233195 | Pg48 | 74.818416 | 55.66222673 | 50.7813389 |
| Pg17 | 500 | 489.2867798 | 489.4316065 | Pg49 | 99.519272 | 167.415806 | 100.5621492 |
| Pg18 | 490.89205 | 399.4800955 | 493.0301498 | Pg50 | 116.0936 | 170.7899652 | 105.6288952 |
| Pg19 | 514.62594 | 511.432115 | 511.337944 | Pg51 | 109.31998 | 167.4168782 | 114.6635218 |
| Pg20 | 525.35426 | 511.3041636 | 550 | Pg52 | 106.01984 | 125.329591 | 133.3378228 |
| Pg21 | 550 | 523.3475504 | 523.296119 | Pg53 | 60.72298 | 60.42436073 | 70.08052756 |
| Pg22 | 548.52995 | 523.2824448 | 527.9275831 | Pg54 | 52.580944 | 79.5458983 | 67.82445444 |
| Pg23 | 550 | 523.2914355 | 549.9999949 | Pg55 | 43.691272 | 56.36733616 | 52.91754938 |
| Pg24 | 521.61373 | 523.3169144 | 536.3000349 | Pg56 | 56.186778 | 66.27287672 | 55.44283757 |
| Pg25 | 522.56351 | 523.306813 | 526.3906729 | Pg57 | 12.98492749 | 10.52629429 | 24.879583 |
| Pg26 | 549.31974 | 523.2792844 | 523.3095942 | Pg58 | 29.50082157 | 13.96655832 | 13.325097 |
| Pg27 | 14.540184 | 10.00833641 | 11.43580483 | Pg59 | 10.22458212 | 10.24231125 | 12.523674 |
| Pg28 | 10.098278 | 10.00401295 | 11.56317354 | Pg60 | 13.31866365 | 18.41685747 | 12.214388 |
| Pg29 | 10.909877 | 10.02394824 | 10.00000666 | Pg61 | 37.73118881 | 46.39822013 | 58.063135 |
| Pg30 | 96.999939 | 96.95242919 | 89.71504585 | Pg62 | 55.41904721 | 77.24184524 | 47.952259 |
| Pg31 | 180.39147 | 181.241749 | 188.3825144 | Pg63 | 38.52470481 | 53.47372752 | 35.709818 |
| Pg32 | 189.8298 | 189.9953502 | 189.9818368 | Pg64 | 58.3696072 | 57.21827797 | 45.08951 |
| | | | | (b) | | | |
| Unit | HBA | JSA | HBJSA | Unit | HBA | JSA | HBJSA |
| Hg41 | 130.3604684 | 133.6418007 | 121.29 | Hg65 | 397.9644411 | 347.8697453 | 409.80933 |
| Hg42 | 130.2202223 | 140.381546 | 136.07204 | Hg66 | 394.0861183 | 349.5293932 | 397.46604 |
| Hg43 | 123.5432156 | 109.2841022 | 134.528 | Hg67 | 400.2039596 | 349.7318595 | 413.03126 |
| Hg44 | 134.1352577 | 118.4055739 | 125.13589 | Hg68 | 401.4451605 | 344.0636931 | 366.65675 |
| Hg45 | 77.30525496 | 121.3009135 | 83.04628 | Hg69 | 59.9999999 | 59.99488855 | 59.953414 |
| Hg46 | 78.02176698 | 82.83206972 | 78.035694 | Hg70 | 59.3632427 | 59.97703456 | 59.988637 |
| Hg47 | 107.1826963 | 106.7556784 | 88.175722 | Hg71 | 59.86117224 | 59.99220729 | 59.466048 |
| Hg48 | 105.056327 | 88.52055969 | 84.30425 | Hg72 | 60 | 59.81462736 | 59.836488 |
| Hg49 | 115.192982 | 153.2842745 | 113.88524 | Hg73 | 58.88396654 | 59.98307532 | 60 |
| Hg50 | 124.3738069 | 155.1483245 | 118.16811 | Hg74 | 59.5417115 | 59.99915877 | 60 |
| Hg51 | 120.69336 | 153.2958557 | 123.68397 | Hg75 | 59.81273203 | 59.90067694 | 60 |
| Hg52 | 118.7925953 | 129.3907541 | 133.9953 | Hg76 | 60 | 59.98083928 | 59.999977 |
| Hg53 | 92.88987192 | 92.63077153 | 100.95165 | Hg77 | 120 | 119.9950741 | 119.99872 |
| Hg54 | 85.7914299 | 109.1376617 | 99.009779 | Hg78 | 120 | 119.9939719 | 117.44199 |
| Hg55 | 78.18666618 | 89.12490012 | 86.055856 | Hg79 | 119.9528822 | 119.9953091 | 119.99999 |
| Hg56 | 88.97404025 | 97.67717242 | 88.298905 | Hg80 | 120 | 119.9308343 | 120 |
| Hg57 | 41.27968283 | 40.22573511 | 46.369847 | Hg81 | 119.9999728 | 119.9583146 | 119.99954 |
| Hg58 | 48.35748487 | 41.69903249 | 41.415021 | Hg82 | 119.4423968 | 119.9958398 | 120 |
| Hg59 | 40.09624275 | 40.10312819 | 41.063184 | Hg83 | 119.999888 | 119.9968772 | 116.18205 |
| Hg60 | 41.2298315 | 43.60456849 | 40.943192 | Hg84 | 111.9564926 | 119.9957398 | 119.08122 |
| Hg61 | 21.18412326 | 25.17541074 | 29.921693 | Sum (Pg) | 12702 | 12701 | 12704 |
| Hg62 | 27.1688682 | 39.19892893 | 25.868861 | Sum (Hg) | 5000 | 5000 | 5000 |
| Hg63 | 21.60258212 | 28.38461052 | 19.393777 | WFC (USD) | 289,822.392 | 290,323.818 | 288,820.7 |
| Hg64 | 25.84708597 | 30.09746698 | 21.476288 | | | | |

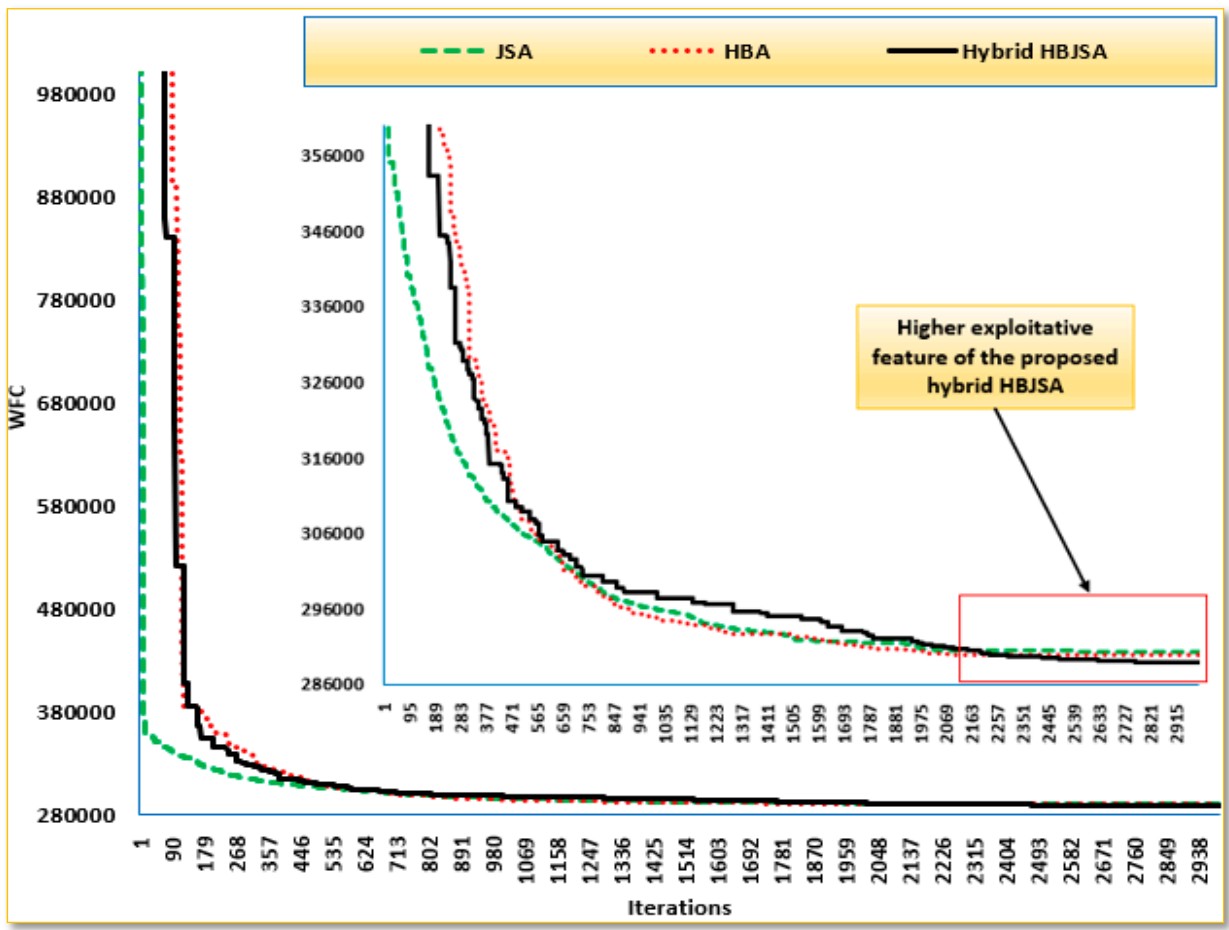

**Figure 7.** Convergence rates of the proposed HBJSA versus the HBA and JSA for the 84-unit system of CHP economic dispatch.

**Table 6.** Comparison between HBA, JSA, and HBJSA with respect to reported techniques for the 84-unit system of CHP economic dispatch.

| Optimizer | WFC (USD) | Rank |
|---|---|---|
| WOA [20] | 290,123.97 | 4 |
| SDA [34] | 292,788.5 | 7 |
| MPA [42] | 294,717.7 | 8 |
| IMPA [42] | 289,903.8 | 3 |
| MRFO [28] | 291,225.6 | 6 |
| HBA | 289,822.4 | 2 |
| JSA | 290,323.8 | 5 |
| Proposed hybrid HBJSA | 288,820.7 | 1 |

### 4.4. Simulation Results of the 96-Unit Test System

The data for the obtained system are mentioned in [20], which illustrates that 12700 MW and 5000 MWth are the load demand and heat demand, respectively, and it has 20 heat units, 52 thermal units, and 24 CHP units. The standard HBA, standard JSA, and proposed HBJSA are applied to this test system, and the corresponding MW, MWth for each unit, and WFC are demonstrated in Table 7. It can be manifested that the proposed HBJSA provides the optimal solution for WFC minimization, which accounts for USD 234,836.04, while the standard HBA and the standard JSA account for USD 235,102.65 and USD 235,277.05, respectively.

**Table 7.** Comparison between HBA, JSA, and the proposed HBJSA for the 96-unit test system of CHP economic dispatch problem. (a) Power outputs from power only and CHP units. (b) Heat outputs from CHP and heat-only units.

|  |  |  | (a) |  |  |  |  |
|---|---|---|---|---|---|---|---|
| Unit | HBA | JSA | HBJSA | Unit | HBA | JSA | HBJSA |
| Pg1 | 537.254715 | 628.137415 | 448.717297 | Pg39 | 92.65458803 | 92.39186824 | 92.49373328 |
| Pg2 | 341.5738074 | 224.3852587 | 299.2517474 | Pg40 | 448.3942915 | 448.8085979 | 442.7202304 |
| Pg3 | 151.1408777 | 224.523037 | 224.1274655 | Pg41 | 297.8196029 | 150.3797127 | 299.2207761 |
| Pg4 | 109.0977628 | 109.8640106 | 111.2441868 | Pg42 | 146.2329552 | 299.2034647 | 299.2279809 |
| Pg5 | 64.10866612 | 109.8714738 | 62.37611714 | Pg43 | 110.6346294 | 111.2974761 | 160.3281579 |
| Pg6 | 110.0480942 | 159.7649454 | 179.2961575 | Pg44 | 161.7767837 | 110.3086384 | 60.04191249 |
| Pg7 | 94.33638081 | 159.7332315 | 99.67889937 | Pg45 | 61.66518066 | 159.5162729 | 60.0000011 |
| Pg8 | 60.00000879 | 110.0519045 | 117.6713057 | Pg46 | 108.9737886 | 109.8987507 | 160.4436803 |
| Pg9 | 108.2875293 | 110.3981768 | 156.6263679 | Pg47 | 110.4947542 | 109.9034171 | 60.69590696 |
| Pg10 | 115.4310336 | 110.6273109 | 78.02377234 | Pg48 | 109.9050818 | 110.2078249 | 115.9607595 |
| Pg11 | 49.01342421 | 77.48642225 | 78.94705246 | Pg49 | 42.02833221 | 77.39752348 | 40.52730593 |
| Pg12 | 92.02655803 | 92.39805239 | 92.1778521 | Pg50 | 72.25178996 | 77.33236811 | 76.26902551 |
| Pg13 | 55.18840658 | 92.42912952 | 88.05166676 | Pg51 | 93.0608812 | 92.76312061 | 86.46078517 |
| Pg14 | 360.9754061 | 358.9389673 | 538.5168446 | Pg52 | 92.57317097 | 93.12310103 | 82.99448914 |
| Pg15 | 299.3860657 | 224.471123 | 224.6561836 | Pg53 | 104.4403419 | 104.9831519 | 93.34146328 |
| Pg16 | 359.9221971 | 227.8984155 | 149.8696746 | Pg54 | 47.60359863 | 45.13159323 | 52.66559089 |
| Pg17 | 159.7120077 | 109.9684494 | 60 | Pg55 | 88.44497134 | 100.2247131 | 106.2115994 |
| Pg18 | 109.3602599 | 109.9616018 | 60 | Pg56 | 50.14389995 | 51.71950188 | 65.28739078 |
| Pg19 | 110.4137284 | 109.8707063 | 154.0137736 | Pg57 | 12.03402877 | 19.00938769 | 10.16852035 |
| Pg20 | 101.8946306 | 110.4372778 | 109.7933677 | Pg58 | 45.6270289 | 56.30321085 | 37.30026155 |
| Pg21 | 109.8648707 | 109.96827 | 60 | Pg59 | 91.41295489 | 125.4849027 | 110.4459566 |
| Pg22 | 179.4592083 | 109.861271 | 159.8078912 | Pg60 | 51.86266145 | 51.90216163 | 56.35148968 |
| Pg23 | 40.13034648 | 114.7915897 | 40.00785124 | Pg61 | 110.2162351 | 91.18311611 | 93.67867133 |
| Pg24 | 77.22962911 | 77.44476922 | 40.00084287 | Pg62 | 47.62594622 | 54.14716313 | 42.99511003 |
| Pg25 | 66.61283247 | 92.41354325 | 92.2948894 | Pg63 | 18.272772 | 34.655934 | 20.975389 |
| Pg26 | 91.02277492 | 92.44381223 | 94.51903986 | Pg64 | 45.155884 | 75.627888 | 35.029188 |
| Pg27 | 359.4498313 | 359.4658937 | 629.8475952 | Pg65 | 88.829361 | 148.19343 | 107.83001 |
| Pg28 | 299.4259674 | 149.5651638 | 151.507932 | Pg66 | 43.74986 | 75.326734 | 51.732368 |
| Pg29 | 289.8301184 | 224.7535524 | 359.9646486 | Pg67 | 95.340679 | 100.58341 | 87.64695 |
| Pg30 | 161.92495 | 109.9432458 | 109.8286727 | Pg68 | 67.297804 | 42.252459 | 52.983829 |
| Pg31 | 107.7669533 | 109.7440166 | 60.38342883 | Pg69 | 11.638244 | 17.202374 | 10.380566 |
| Pg32 | 159.4640725 | 109.899657 | 109.8614158 | Pg70 | 35.028678 | 47.915618 | 37.11649 |
| Pg33 | 162.5783378 | 112.3437083 | 160.9958816 | Pg71 | 87.384128 | 99.938915 | 110.78998 |
| Pg34 | 159.9001838 | 118.7834711 | 159.7427634 | Pg72 | 53.523651 | 67.954604 | 44.094744 |
| Pg35 | 60.03756814 | 109.9579597 | 109.7936225 | Pg73 | 106.28473 | 88.257959 | 99.595259 |
| Pg36 | 113.7051781 | 114.8045459 | 120 | Pg74 | 65.916093 | 53.319524 | 62.695199 |
| Pg37 | 114.3819608 | 77.49926496 | 40.00494973 | Pg75 | 12.449246 | 11.427801 | 10.796383 |
| Pg38 | 94.29271341 | 92.63059511 | 86.65823591 | Pg76 | 35.002312 | 47.191073 | 44.243458 |

|  |  |  | (b) |  |  |  |  |
|---|---|---|---|---|---|---|---|
| Unit | HBA | JSA | HBJSA | Unit | HBA | JSA | HBJSA |
| Hg53 | 117.8187 | 118.25717 | 111.65705 | Hg77 | 385.57459 | 400.14747 | 390.26983 |
| Hg54 | 81.187872 | 79.425882 | 85.919458 | Hg78 | 59.99772 | 59.999812 | 56.453332 |
| Hg55 | 108.9226 | 115.58344 | 118.85523 | Hg79 | 60 | 59.922567 | 59.999913 |
| Hg56 | 83.024668 | 85.11551 | 96.801938 | Hg80 | 118.82035 | 119.99872 | 119.42771 |
| Hg57 | 40.490709 | 43.852427 | 40.05269 | Hg81 | 119.99892 | 119.97171 | 119.96065 |
| Hg58 | 24.658925 | 29.679309 | 15.297858 | Hg82 | 438.22049 | 402.77116 | 436.82909 |
| Hg59 | 109.94761 | 129.76254 | 116.53145 | Hg83 | 60 | 59.993989 | 59.982729 |
| Hg60 | 83.140535 | 85.274518 | 89.077417 | Hg84 | 59.858041 | 59.997954 | 59.888563 |
| Hg61 | 120.77066 | 110.51193 | 111.89757 | Hg85 | 119.95149 | 119.99343 | 119.76019 |
| Hg62 | 81.226938 | 87.148267 | 77.525341 | Hg86 | 118.46015 | 119.99908 | 119.9878 |
| Hg63 | 39.67663 | 50.561873 | 44.68415 | Hg87 | 450.50341 | 399.42417 | 429.43143 |
| Hg64 | 23.984424 | 38.46521 | 19.947464 | Hg88 | 59.784083 | 59.977283 | 60 |
| Hg65 | 108.20515 | 142.50587 | 119.75308 | Hg89 | 59.830588 | 59.991776 | 59.830142 |
| Hg66 | 77.96514 | 105.49604 | 85.12105 | Hg90 | 119.99947 | 119.9983 | 120 |
| Hg67 | 112.78318 | 115.78643 | 108.31842 | Hg91 | 118.32769 | 119.99419 | 119.89929 |
| Hg68 | 98.469051 | 76.943631 | 85.900861 | Hg92 | 451.81766 | 398.46975 | 449.27757 |

**Table 7.** *Cont.*

**(b)**

| Unit | HBA | JSA | HBJSA | Unit | HBA | JSA | HBJSA |
|------|-----|-----|-------|------|-----|-----|-------|
| Hg69 | 40.664318 | 43.086924 | 40.158344 | Hg93 | 60 | 59.998636 | 59.693249 |
| Hg70 | 18.721909 | 25.870776 | 20.947021 | Hg94 | 59.916361 | 59.985181 | 59.957398 |
| Hg71 | 106.59216 | 115.42154 | 121.3902 | Hg95 | 119.99988 | 119.98799 | 119.99901 |
| Hg72 | 86.491633 | 99.131229 | 78.521522 | Hg96 | 119.83914 | 119.99335 | 119.9259 |
| Hg73 | 118.09871 | 108.86068 | 115.18485 | Sum (Pg) | 9400 | 9400 | 9400 |
| Hg74 | 97.014286 | 86.491059 | 94.56892 | Sum (Hg) | 5000 | 5000 | 5000 |
| Hg75 | 40.89038 | 40.610943 | 37.166946 | WFC | 235,102.65 | 235,277.05 | 234,836.04 |
| Hg76 | 18.353784 | 25.540248 | 24.147382 | | | | |

Moreover, convergence characteristics of the proposed HBJSA versus the standard HBA and the standard JSA for the 96-unit test system of the CHP economic dispatch problem are depicted in Figure 8. From this figure, the proposed hybrid HBJSA is capable of improving the solution quality compared to the standard HBA and the standard JSA. At the last 1000 iterations, the proposed hybrid HBJSA provides a higher exploitative feature and, finally, reaches the lowest WFC of USD 234,836.04. Additionally, the standard HBA, the standard JSA, and the proposed HBJSA effectively achieve all constraints with 100% accuracy, as illustrated in Table 7.

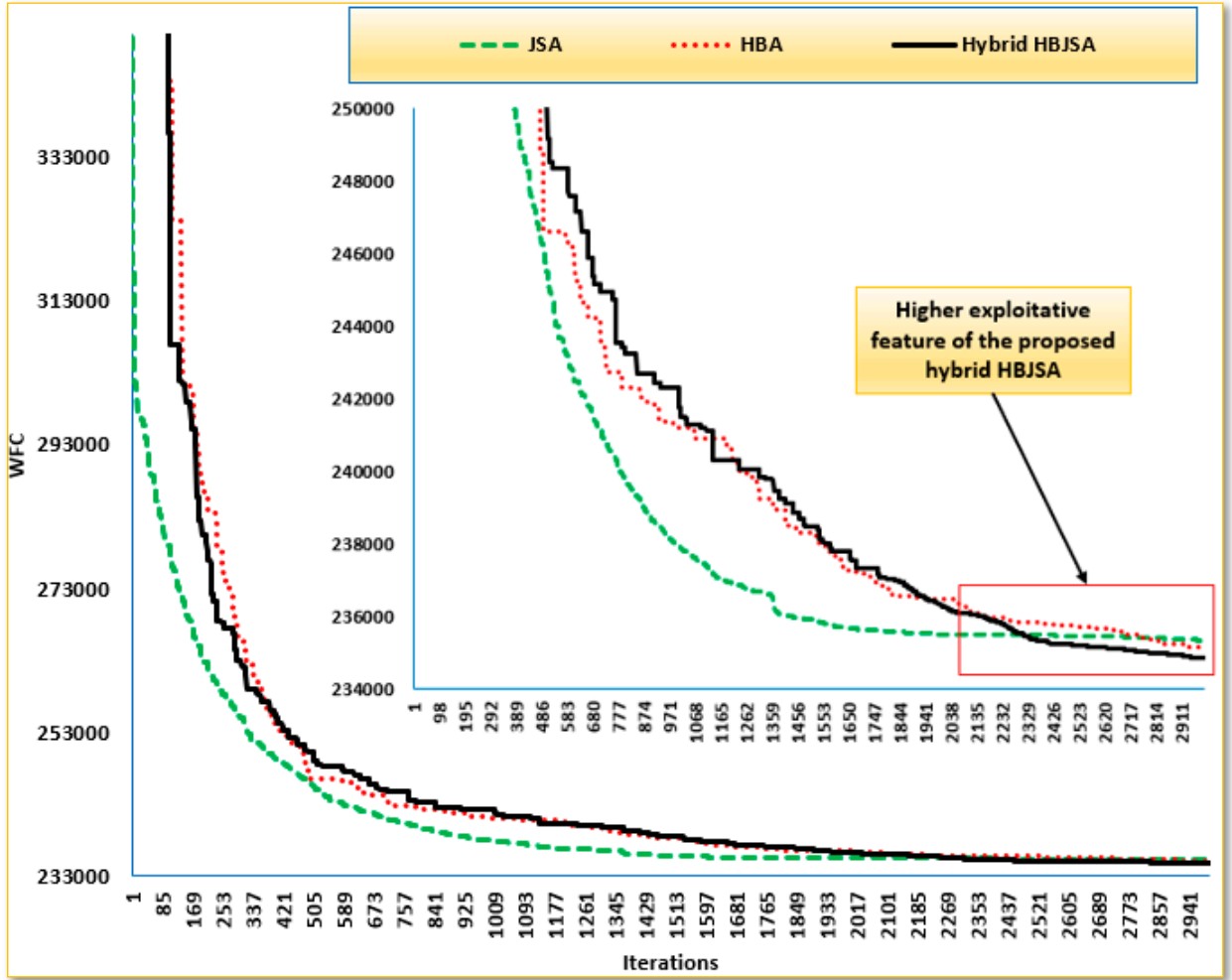

**Figure 8.** Convergence characteristics of the proposed HBJSA versus the HBA and JSA for the 96-unit system of CHP economic dispatch.

In addition, a comparison study between the standard HBA, JSA, and the proposed HBJSA is conducted in Table 8 for the 96-unit system of CHP economic dispatch with respect to reported techniques such as WVO-PSO [55], WOA [20], MPA [42], IMPA [42], MRFO [29], and SDA [34]. In this table, the ranking order is evaluated in ascending order based on the minimum WFC. From this table, the proposed hybrid HBJSA achieves the first rank with the lowest WFC. On the other side, the standard HBA occupies the second rank, while the standard JSA occupies the fourth rank. Additionally, this table demonstrates that the proposed HBJSA overwhelmed the standard HBA and the standard JSA and reported recent techniques for achieving minimum WFC.

**Table 8.** Comparison between HBA, JSA, and HBJSA with respect to reported techniques for the 96-unit test system of CHP economic dispatch problem.

| Optimizer | Sum (Pg) | Sum (Hg) | WFC (USD) | Rank |
|---|---|---|---|---|
| WOA [20] | 9400.033 | 5000 | 236,699.15 | 8 |
| WVO-PSO [55] | 9399.99 | 4999.99 | 238,005.79 | 9 |
| SDA [34] | 9400 | 5000 | 236,185.18 | 6 |
| MRFO [29] | 9400 | 5000 | 235,541.4 | 5 |
| MPA [42] | 9400 | 5000 | 236,283.1 | 7 |
| IMPA [42] | 9400 | 5000 | 235,260.3 | 3 |
| HBA | 9400 | 5000 | 235,102.65 | 2 |
| JSA | 9400 | 5000 | 235,277.05 | 4 |
| Proposed hybrid HBJSA | 9400 | 5000 | 234,836.04 | 1 |

### 4.5. Statistical Assessment of HBA, JSA, and Proposed Hybrid HBJSA for CHP Economic Dispatch

For all test systems, the proposed hybrid HBJSA, HBA, and JSA are run several times, and the corresponding whiskers box plots are drawn in Figure 9. For the 24-unit system, as shown in Figure 9a, the proposed hybrid HBJSA outperforms HBA and JSA in finding the lower minimum, average, and maximum WFC values. The proposed hybrid HBJSA achieves minimum, average, and maximum WFC values of USD 57,968.539, USD 58,103.95, and USD 58,293.6, respectively. On the other side, the HBA achieves minimum, average, and maximum WFC values of USD 57,994.51, USD 58,111.3, and USD 58,309.416, respectively, whereas the JSA obtains counterparts of USD 58,739.524, USD 58,968.565, and USD 59,125.33, respectively.

For the 48-unit system, as shown in Figure 9b, the proposed hybrid HBJSA outperforms HBA and JSA in finding the lowest minimum WFC value of USD 116,140.335. Compared to the HBA, the proposed hybrid HBJSA obtains lower maximum WFC values of USD 117,848.43 where the HBA obtains USD 117,980.55, while both acquire comparable WFC values of USD 116,952.6 and USD 116,946.22 for the proposed hybrid HBJSA and HBA, respectively. Compared to the JSA, the proposed hybrid HBJSA presents great superiority, since the JSA obtains minimum, average, and maximum WFC values of USD 117,365.09, USD 117,911.105, and USD 118,456.98, respectively.

For the 84-unit system, as shown in Figure 9c, the proposed hybrid HBJSA outperforms HBA and JSA in finding the lower minimum, average, and maximum WFC values. The proposed hybrid HBJSA achieves minimum, average, and maximum WFC values of USD 288,820.68, USD 289,813.827, and USD 291,251.73, respectively. On the other side, the HBA achieves minimum, average, and maximum WFC values of USD 289,822.392, USD 290,891.01, and USD 292,342.51, respectively, whereas the JSA obtains counterparts of USD 290,323.82, USD 292,366.86, and USD 293,747.44, respectively.

For the 96-unit system, as shown in Figure 9d, the proposed hybrid HBJSA outperforms HBA and JSA in finding the lower minimum, average, and maximum WFC values. The proposed hybrid HBJSA achieves minimum, average, and maximum WFC values of USD 234,836.0389, USD 235,646.129, and USD 235,967.06, respectively. On the other side, the HBA achieves minimum, average, and maximum WFC values of USD 235,102.65, USD 2,356,921.613, and USD 239,119.46, respectively, whereas the JSA obtains counterparts of USD 235,277.05, USD 236,688.76, and USD 237,940.189, respectively.

All these comparative assessments illustrate the high stability and robustness of the proposed HBJSA in finding the lowest minimum, average, and maximum WFC value compared with the HBA and JSA.

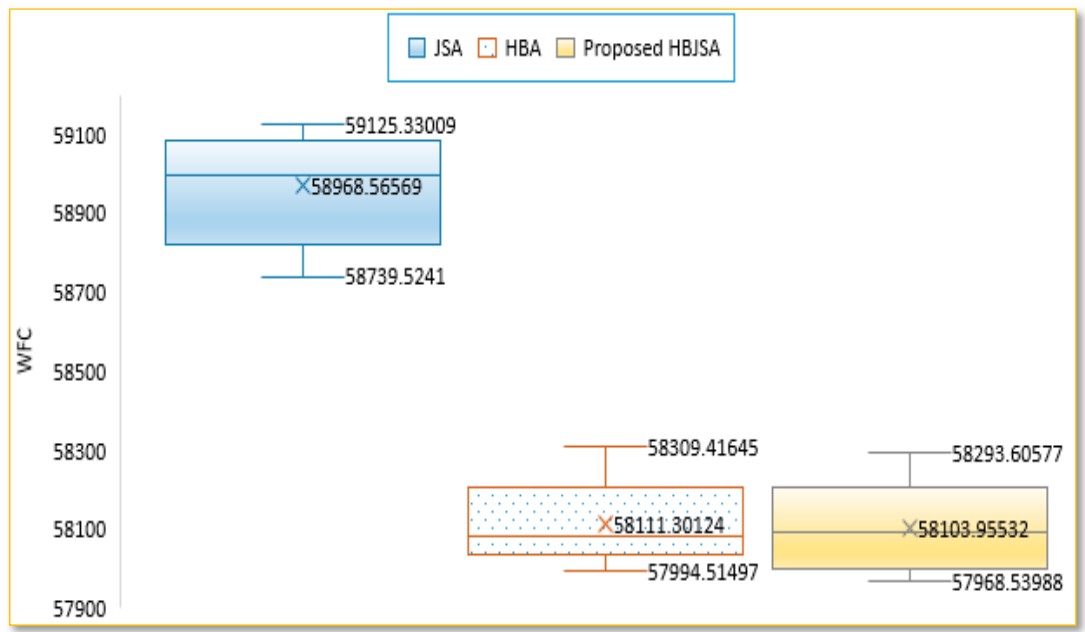

(**a**) 24-unit test system.

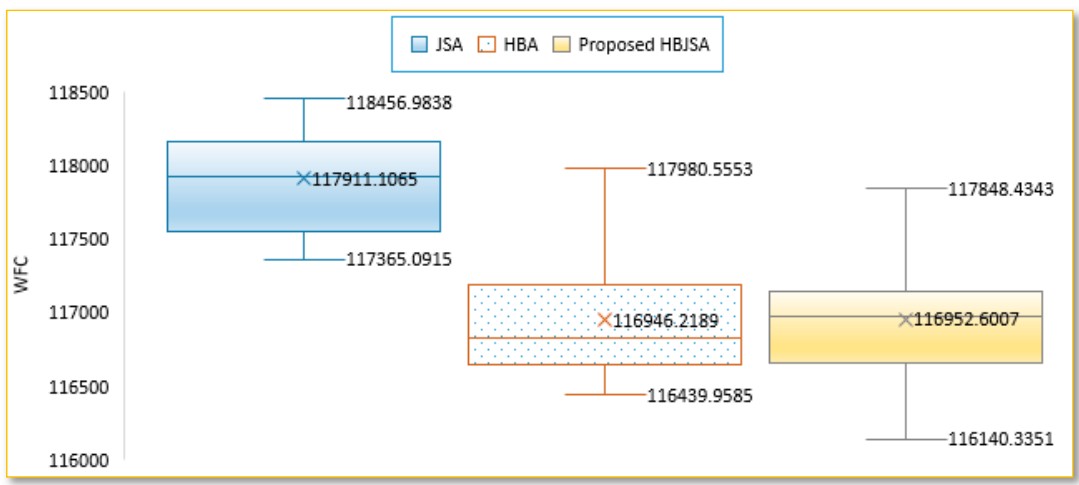

(**b**) 48-unit test system.

**Figure 9.** *Cont.*

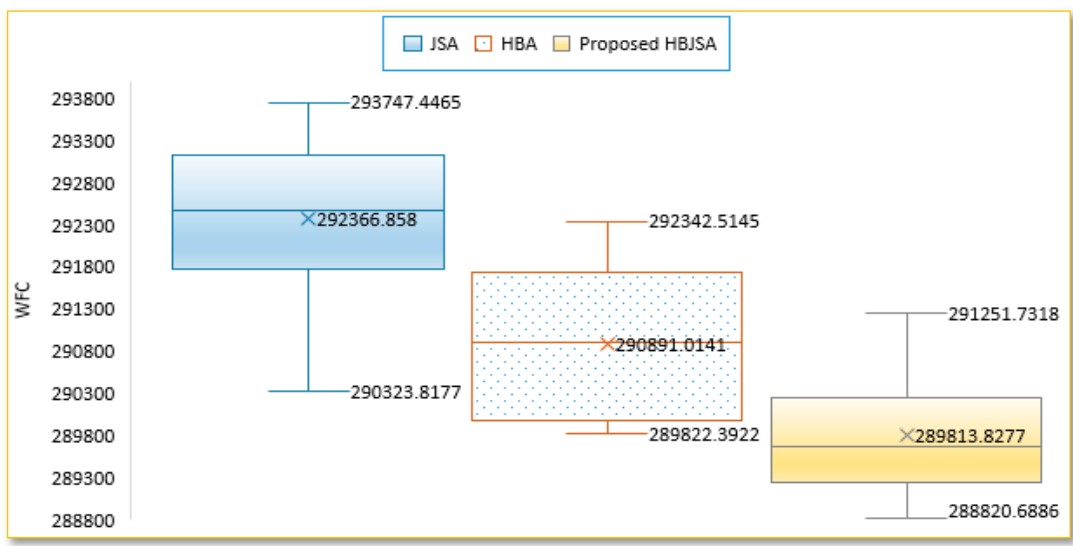

(**c**) 84-unit test system.

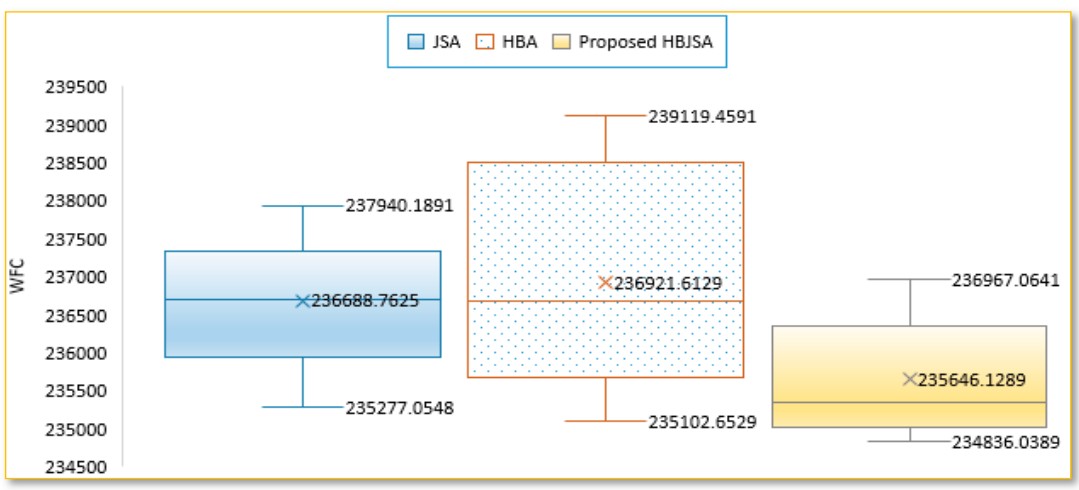

(**d**) 96-unit test system.

**Figure 9.** Whiskers box plot for the proposed HBJSA versus HBA and JSA for solving the CHP economic dispatch problem.

From these implementations, the practical use of the HJBSA for a larger scale as 84-unit and 96-unit test systems do not require cloud solutions. It requires the input data of the system as follows:

- The data of the power and heat loads.
- The data of the power limits of power-only units.
- The data of the heat production limits of heat-only units.
- The data of the power and heat characteristics curves of the CHP units.

## 5. Conclusions

In this paper, an innovative hybrid heap-based and jellyfish search algorithm (HBJSA) is presented for solving the CHP economic dispatch problem. The proposed hybrid heap-based and jellyfish search algorithm (HBJSA) combines the benefits of the standard HBA and standard JSA. Compared with standard HBA and standard JSA, the proposed HBJSA uses an adjustment mechanism in order to support the explorative and exploitative characteristics. In the proposed HBJSA, an adjustment mechanism has been constructed to boost the explorative feature at the start of iterations by enhancing the generated solutions

via HBA. Furthermore, towards the conclusion of iterations, it augments and enhances the exploitative feature by growing the generated solutions via JSA. Besides, the HBA, JSA, and the proposed HBJSA have been utilized to solve the complex CHP economic dispatch problems with hard constraints, which are the feasible operating area of CHP units and valve-point effects. They are applied on two medium systems, which are 24-unit and 48-unit systems, and two large systems, which are 84-unit and 96-unit systems.

The major contributions of this paper are:

- A novel hybrid HBJSA is proposed, for the first time, in order to enhance the performance of the standard HBA and JSA for solving the CHP economic dispatch problem.
- Significant improvements via the proposed HBJSA are achieved in terms of the solution quality with high exploitative convergence characteristics for all systems studied.
- High superiority of the proposed hybrid HBJSA has been satisfied compared with several competitive algorithms in the literature.
- High robustness and stability of the proposed hybrid HBJSA with respect to standard HBA and JSA in finding the lowest minimum, average, and maximum WFC objectives.

**Author Contributions:** A.G.: conceptualization, methodology, writing—original draft preparation; A.E.: validation, writing—original draft; A.S.: software, data curation, writing—original draft preparation, visualization, investigation. R.E.-S.: supervision, validation, revision; corresponding author E.E.: writing—reviewing and editing, funding. All authors have read and agreed to the published version of the manuscript.

**Funding:** This research was funded by Taif University, grant number TURSP-2020/86.

**Institutional Review Board Statement:** Not applicable.

**Informed Consent Statement:** Not applicable.

**Data Availability Statement:** Not applicable.

**Acknowledgments:** This work was supported by Taif University Researchers Supporting Project number (TURSP-2020/86), Taif University, Taif, Saudi Arabia.

**Conflicts of Interest:** The authors declare no conflict of interest.

## Nomenclature

| | |
|---|---|
| $a_k$, $b_k$, $c_k$, $d_k$, $e_k$ and $f_k$ | Cost coefficients of the $k$th unit |
| $a_j$; $b_j$ and $c_j$ | Cost coefficients of $j$th heat plant |
| $a_i$, $b_i$ and $c_i$ | Cost coefficients of $i$th power plant |
| WFC | Whole fuel cost |
| $C_i\left(P_i^{pp}\right)$ | Fuel cost of power unit $i$ |
| $C_j\left(H_j^{hp}\right)$ | Fuel cost of $j$th heat plant |
| $C_k\left(P_k^{cp}, H_k^{cp}\right)$ | The operational cost of $k$th cogeneration unit |
| $P_k^{cp_{Limit}}\left(H_k^{cp}\right)$ | Power bound for the set heat-output of cogenerator $(k)$ |
| $BI$ | Binary coefficient |
| $N_{pp}$ | Number of power-only plants |
| $H_d$ | System heat load |
| $N_{cp}$ | Number of cogenerators |
| $N_{hp}$ | Number of heat-only units |
| $P_d$ | System power load |
| $H^c$ | Heat output of CHP |
| $P^c$ | The power output of CHP |
| $\psi_v$ | Penalty coefficient |
| $\lambda_i$ and $\rho_i$ | Valve-point cost coefficients |
| CRH | Corporate rank hierarchy |
| $t$ | Current iteration |

| | |
|---|---|
| $k$ | $k$th vector component |
| \| \| | Absolute value |
| $(2r - 1)$ | $k$th component of vector $\overrightarrow{\lambda}$ |
| $r$ | Random number from the range [0,1] |
| $f$ | Fitness of the search agent |
| $p$ | Produced randomly number [0,1] |
| $C$ | User-defined parameter which its unit is (iteration) |
| $CO_o$ | Constant equals 0.5 |
| $X_i$ | $i$th jellyfish logistic chaotic value |
| TC | Time control |
| $CF(t)$ | Time control function |
| $t$ | Iteration number |
| $T^{max}$ | Maximum iterations' numbers |
| $\mu$ | Mean for all jellyfish locations in the swarm |
| $P_0$ | The initial jellyfish population, $P_0 \in (0,1), P_0 \notin \{0.0, 0.25, 0.75, 0.5, 1.0\}$. |
| R | A random number from [0–1] |
| $X^*$ | Best location of currant jellyfish |
| f | Objective function |
| $U_b$ | Search spaces upper limit |
| $L_b$ | Search spaces lower limit |
| $X_{i,d}$ | $i$th jellyfish location in $d$th dimension |

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
