# Peer review of "An Innovative Hybrid Heap-Based and Jellyfish Search Algorithm for Combined Heat and Power Economic Dispatch in Electrical Grids"

_mathematics, doi:10.3390/math9172053_

Round 1

Reviewer 1 Report

Dear authors,

congratulations for your work. Sometimes it is difficult to go through all the concepts involved but it is reasonably well done.

I recommend to update your paper as follows:

  • Line 27: Plants should be written as "plants" I guess.
  • Line 133: contributions should be listed at the end of the paper, not before. Please make this change.
  • Line 183: the units of parameter C are missing.
  • Figures 5, 6, 7 and 8: I guess that the term in the leyend is "exploitative". Please confirm and/or update.

In general terms, I miss at least some single line diagrams where the entire power systems studied could be seen.

To have a clear idea of the brilliance of their work, you should include a figure/s that shows the current situation of the electrical power systems studied versus the situation they would have with their new hybrid algorithm put into service. The work is worth including such figures that allow readers to clearly see the benefits of your research.

I would like to have my recommendation included in your final version.

Best wishes.

Author Response

Response to Reviewer # 1

Dear authors,

Congratulations for your work. Sometimes it is difficult to go through all the concepts involved but it is reasonably well done.

Response 1: I express with a profound sense of gratitude and sincere thanks to you for your words and I hope the extended modifications find your satisfaction

I recommend to update your paper as follows:

  • Line 27: Plants should be written as "plants" I guess.

Response: Ok. It is modified.

  • Line 133: contributions should be listed at the end of the paper, not before. Please make this change.

Response: Ok. It is transferred to the end of the paper.

  • Line 183: the units of parameter C are missing.

Response: Ok. C is a user-defined parameter which its unit is (iteration). It is defined in the nomenclature as follows:.

C

user-defined parameter which its unit is (iteration)

  • Figures 5, 6, 7 and 8: I guess that the term in the legend is "exploitative". Please confirm and/or update.

Response: Ok. Figures 5, 6, 7 and 8 are modified. The term in the legend is modified to "exploitative"

In general terms, I miss at least some single line diagrams where the entire power systems studied could be seen.  To have a clear idea of the brilliance of their work, you should include a figure/s that shows the current situation of the electrical power systems studied versus the situation they would have with their new hybrid algorithm put into service. The work is worth including such figures that allow readers to clearly see the benefits of your research.

The proposed algorithm can be applied for the level of transmission, distribution network or households. The general form of the combined heat and power dispatch is added to the paper in Fig. 1 where it shows a single line diagram of the 24-unit test system for CHP economic dispatch problem, as an example. As shown, different sources of the CHP, heat only and power only units supply power and heat to satisfy the power and heat demands. Heat production and power output balance means two items. Firstly, the summation of power generation equals the total power load. Secondly, the summation of heat generation equals the total heat load.

Fig. 1: single line diagram of the 24-unit test system for CHP economic dispatch problem

Reviewer 2 Report

  • What exactly do the authors mean by “power out balance” (Line 133)? Is the proposed algorithm intended for the level of distribution network operators (DSO) or maybe households? A comment would also be required on, for example, the relationship between the results obtained from the HJBSA algorithm and the peak demand phenomenon. Is there any correlation in this case?
  • The paper contains considerations of simulation tests. A commentary on the practical use of the HJBSA algorithm would be required. Would an algorithm for a larger scale with more data require cloud solutions?
  • There is also a lack of information which IT tools were used by the authors during the implementation of the HJBSA algorithm. Was it an original implementation or maybe ready dedicated libraries were used?
  • The rankings of the individual algorithms are based on the WFC values. A comment on the variable number of significant digits for different algorithms, eg in Table 5, would be required. Did the applied objects for storing data with the given precision not influence the obtained WFC values? I mean using the BigDecimal class when implementing a Java program in dedicated applications.
  • The data in Table 5 are hardly legible. A modification would be required.
  • It would be advisable to extend chapter 3.3 Proposed Hybrid HBJSA with implementation details, e.g. based on a simplified example.
  • In chapter 0 Nomenclature, the meaning of the following is ambiguous:
  • C as user defined parameter or as Total production cost,
  • Maxiter maximum iterations' numbers and T Total iterations ’number

Author Response

Response to Reviewer # 2

Comments and Suggestions for Authors

  • What exactly do the authors mean by “power out balance” (Line 133)?

Response: In Line 133, Heat production and power output balance means two items:

  1. The summation of power generation equals the total power load.
  2. The summation of heat generation equals the total heat load.

This point is added to section 2 of problem Formulation

Is the proposed algorithm intended for the level of distribution network operators (DSO) or maybe households?

Response: The proposed algorithm can be applied for the level of transmission, distribution network or households. The general form of the combined heat and power dispatch is added to the paper in Fig. 1 where it shows a single line diagram of the 24-unit test system for CHP economic dispatch problem, as an example. As shown, different sources of the CHP, heat only and power only units supply power and heat to satisfy the power and heat demands. Heat production and power output balance means two items. Firstly, the summation of power generation equals the total power load. Secondly, the summation of heat generation equals the total heat load.

Fig. 1: single line diagram of the 24-unit test system for CHP economic dispatch problem

A comment would also be required on, for example, the relationship between the results obtained from the HJBSA algorithm and the peak demand phenomenon. Is there any correlation in this case?

Response: The CHP economic dispatch problem is a problem where the loads are varied through the day and so it a management problem that the loads are always varied. Therefore, the proposed algorithm is applied in the paper to specified power and heat loading values, at a certain hour, for different four systems. Each system has specified power and heat loading. For example, for the 24-unit system in Fig. 1, the power load is 2350 MW while the heat load is 1250 MWth. Since the validity of the proposed algorithm is demonstrated for different cases, it also has the validity for different loading conditions. Nevertheless, the considered loadings are the peak loadings that are reported in previous papers which are included in the comparisons in the paper.

  • The paper contains considerations of simulation tests. A commentary on the practical use of the HJBSA algorithm would be required. Would an algorithm for a larger scale with more data require cloud solutions?

Response: Ok. The practical use of the HJBSA algorithm for a larger scale as 84-unit and 96-unit test systems don’t require cloud solutions. It requires the input data of the system as follows:

  • The data of the power and heat loads.
  • The data of the power limits of power-only units
  • The data of the heat production limits of heat-only units
  • The data of the power and heat characteristics curves of the CHP units

This point is clarified in Section

  • There is also a lack of information which IT tools were used by the authors during the implementation of the HJBSA algorithm. Was it an original implementation or maybe ready dedicated libraries were used?

Response: Ok. The MatlabR2017b is used to perform the simulations on a system with 8 GB of RAM and Intel(R) Core (TM) i7-7200U CPU (2.5 GHz). This point is clarified now in the simulation results of Section 4.

  • The rankings of the individual algorithms are based on the WFC values. A comment on the variable number of significant digits for different algorithms, eg in Table 5, would be required. Did the applied objects for storing data with the given precision not influence the obtained WFC values? I mean using the BigDecimal class when implementing a Java program in dedicated applications.

Response: Ok.  In MATLAB tool, the precision is very very high and so the applied objects for storing data with the given precision doesnot influence the obtained WFC values

  • The data in Table 5 are hardly legible. A modification would be required.

Response: Ok. Table 5 has been divided into two parts for simple illustration as follows:

Table 5. Comparison between HBA, JSA and the proposed HBJSA for the 84-unit test system of CHP economic dispatch problem.

Table5(a). Power outputs from power only and CHP units

Table 5(b). Heat outputs from CHP and heat-only units

Similarly, Table 7 has been divided into two parts for simple illustration as follows: Comparison between HBA, JSA and the proposed HBJSA for the 96-unit test system of CHP economic dispatch problem.

Table7(a). Power outputs from power only and CHP units

Table 7(b). Heat outputs from CHP and heat-only units

  • It would be advisable to extend chapter 3.3 Proposed Hybrid HBJSA with implementation details, e.g. based on a simplified example.

Response: Section 3.3 is extended to cover the implementation procedure with the main steps and simplified example as recommended.

  • In chapter 0 Nomenclature, the meaning of the following is ambiguous:

Response: Ok. They are treated.

  • C as user defined parameter or as Total production cost,

Response: Ok. C is a user-defined parameter which its unit is (iteration). WFC is the Whole Fuel Cost.  They are defined in the nomenclature as follows:.

WFC

Whole Fuel Cost

C

user-defined parameter which its unit is (iteration)

  • Maxiter maximum iterations' numbers and T Total iterations ’number

Response: Ok. They are unified to Tmax.

Tmax

maximum iterations' numbers
